# Inactivated vaccine-elicited potent antibodies can broadly neutralize SARS-CoV-2 circulating variants

Yubin Liu [1,2,9], Ziyi Wang[3,9], Xinyu Zhuang[4,9], Shengnan Zhang[5,9], Zhicheng Chen[6,9], Yan Zou[2], Jie Sheng[2], Tianpeng Li[6], Wanbo Tai[2], Jinfang Yu[3], Yanqun Wang[5], Zhaoyong Zhang[5], Yunfeng Chen [2], Liangqin Tong [1], Xi Yu [1], Linjuan Wu[1], Dong Chen[7], Renli Zhang [8], Ningyi Jin[4], Weijun Shen [6]✉, Jincun Zhao [5]✉, Mingyao Tian [4]✉, Xinquan Wang [3]✉ & Gong Cheng [1,2]✉

A full understanding of the inactivated COVID-19 vaccine-mediated antibody responses to SARS-CoV-2 circulating variants will inform vaccine effectiveness and vaccination development strategies. Here, we offer insights into the inactivated vaccine-induced antibody responses after prime-boost vaccination at both the polyclonal and monoclonal levels. We characterized the VDJ sequence of 118 monoclonal antibodies (mAbs) and found that 20 neutralizing mAbs showed varied potency and breadth against a range of variants including XBB.1.5, BQ.1.1, and BN.1. Bispecific antibodies (bsAbs) based on nonoverlapping mAbs exhibited enhanced neutralizing potency and breadth against the most antibody-evasive strains, such as XBB.1.5, BQ.1.1, and BN.1. The passive transfer of mAbs or their bsAb effectively protected female hACE2 transgenic mice from challenge with an infectious Delta or Omicron BA.2 variant. The neutralization mechanisms of these antibodies were determined by structural characterization. Overall, a broad spectrum of potent and distinct neutralizing antibodies can be induced in individuals immunized with the SARS-CoV-2 inactivated vaccine BBIBP-CorV, suggesting the application potential of inactivated vaccines and these antibodies for preventing infection by SARS-CoV-2 circulating variants.

Severe acute respiratory syndrome coronavirus-2 (SARS-CoV-2), the etiological agent of coronavirus disease 2019 (COVID-19), has caused a severe pandemic and public health threat worldwide. Accumulating epidemiological data show that confirmed cases, deaths, and the spread of COVID-19 are still rising[1]. Several vaccines developed with multiple strategies that confer excellent protection efficacy against COVID-19 have been broadly administered under full World Health Organization (WHO) approval[2]. However, continuously emerging SARS-CoV-2 variants frequently escape the neutralizing barrier formed by either natural infection or vaccination, thereby diminishing the effectiveness of the available vaccines and reducing the efficacy of antibody-based therapeutics[3–7]. Polyclonal sera from convalescent or vaccinated individuals show substantially lower neutralizing activity against the pandemic Omicron sublineages than the original strains[8–14]. During periods of Delta or Omicron variant predominance, a third vaccine dose has been highly effective in protecting individuals against severe COVID-19-related outcomes and preventing COVID-19-associated hospitalizations[15,16]. Furthermore, higher levels of binding

A full list of affiliations appears at the end of the paper. ✉e-mail: wshen@szbl.ac.cn; zhaojincun@gird.cn; klwklw@126.com; xinquanwang@mail.tsinghua.edu.cn; gongcheng@mail.tsinghua.edu.cn

and neutralizing antibodies are correlated with a reduced risk of symptomatic infection, and neutralizing antibody levels are highly predictive of immune protection[17,18]. Therefore, a full understanding of the vaccine-mediated antibody response to SARS-CoV-2 and circulating variants is needed for a better evaluation of SARS-CoV-2 vaccine effectiveness.

Inactivated COVID-19 vaccines, which are based on a traditional platform, show high safety and effectiveness and are thus being deployed globally to prevent COVID-19[19–23]. The serum or plasma antibody response to the current commercial inactivated vaccines has been studied extensively[24–28]. However, the profile of the plasma antibody response elicited by inactivated vaccines against all the previously circulating variants of concern (VOCs) (Alpha, Beta, Gamma, and Delta), previously circulating variants of interest (VOIs) (Lambda, Mu, Kappa, Eta, Iota v1, Iota v2, Epsilon, and Zeta), and currently circulating VOCs (BA.1, BA.2, BA.2.12.1, BA.2.75, BA.2.75.2, BN.1, BA.3, BA.4, BA.4.6, BA.5, BF.11, BQ.1, BQ.1.1, XD, XBB, and XBB.1.5) are less well defined. Specifically, the characteristics of the antibody response at the monoclonal level are still poorly defined. Furthermore, the efficacy of vaccine-induced neutralizing antibodies and bispecific antibodies (bsAbs) based on these monoclonal antibodies (mAbs) is also rarely reported. Thus, a longitudinal and comprehensive analysis of the characteristics of antibody responses to inactivated vaccines and a characterization of potent and broadly neutralizing antibodies will be informative to optimize and update vaccine design and immunization strategies as well as therapeutics.

Here, we show the longitudinal humoral response to SARS-CoV-2 in individuals immunized with the BBIBP-CorV inactivated vaccine. A booster vaccine dose can markedly increase plasma binding and neutralizing activity, as well as memory B response, to SARS-CoV-2 and its variants. At the molecular level, IGHV3-30, IGHV3-53, IGKV1-39 and IGLV3-21 are overrepresented in the RBD-binding mAbs, of which 20 mAbs exhibit potent neutralizing activity against SARS-CoV-2, especially antibody 6-2C, which can broadly neutralize all the tested variants. BsAbs generated from 6-2C and nonoverlapping antibodies obtain further increased potency and breadth. Their neutralizing activity was confirmed in female human angiotensin-converting enzyme 2 (hACE2) transgenic mice. We also show their structural basis of binding and neutralization. These findings provide extended information for the efficacy of BBIBP-CorV and suggest the application potential of inactivated vaccines and these antibodies for preventing infection by SARS-CoV-2 circulating variants.

## Results

### Humoral responses to SARS-CoV-2 after BBIBP-CorV vaccination

To characterize the humoral response to the SARS-CoV-2 inactivated vaccine, we analyzed samples from a cohort of 28 healthy volunteers who received the BBIBP-CorV SARS-CoV-2 inactivated vaccine (Supplementary Table 1). Samples were collected over a time course covering before and after immunization to investigate the longitudinal characteristics of the humoral response (Fig. 1a). The plasma IgM and IgG responses against SARS-CoV-2 structural proteins were measured with an enzyme-linked immunosorbent assay (ELISA). Consistent with previous reports, prevaccination samples had detectable antibodies against SARS-CoV-2 structural proteins[29,30]. These antibodies were probably induced by other coronaviruses, particularly seasonally spreading human coronaviruses (HCoVs), and cross-reactive against SARS-CoV-2 spike and nucleocapsid proteins[31,32]. The levels of circulating spike (S)-, receptor-binding domain (RBD)-, S2- and N protein-specific IgM were increased two weeks after the second dose, and a third vaccine dose did not significantly increase the IgM levels compared with those at the same time points after the second vaccination (Supplementary Fig. 1a). The IgG responses to these viral structural proteins were much higher than those of IgM, with strong responses recorded after the two-dose

regimen; a booster dose led to a further 2.3–9.4-fold increase in IgG titers against these antigens (Fig. 1b).

We next determined the plasma neutralizing activity using human immunodeficiency virus-1 (HIV-1) pseudotyped with SARS-CoV-2 S protein. Neutralizing activity against wild-type (WT) SARS-CoV-2 (Wuhan-Hu-1 strain) was detectable in all the vaccinees after the two-dose regimen, but it waned over time. A third dose nonetheless significantly boosted the neutralizing titers by more than 14-fold (Supplementary Fig. 1b). The plasma neutralizing activity against WT SARS-CoV-2 was positively correlated with anti-S and anti-RBD-binding IgG titers determined by ELISA (Supplementary Fig. 1c). Emerging SARS-CoV-2 variants might compromise the efficacy of existing vaccines and therapeutic antibodies[33–38]. We thus assessed the neutralizing activity of the plasma from vaccinated individuals against a panel of 28 pseudotyped SARS-CoV-2 variants (Supplementary Table 2). Consistent with the results for other vaccines, the Omicron variant exhibited substantial immune escape, with only 0–43% of vaccinated individuals having detectable plasma neutralizing activity against its sublineages (Fig. 1c). However, neutralizing titers against the variants were markedly increased after a booster dose of BBIBP-CorV, except for the most antibody-evasive strains, such as BA.2.75.2, BN.1, XBB, and XBB.1.5 (Fig. 1c). Notably, plasma neutralizing activity against Omicron sublineages (BA.1, BA.2, BA.3, BA.4, BA.5) after booster immunization was detectable in more than 81% of the samples, although the percentages declined to 4–58% against the currently prevalent subvariants (BQ.1, BQ.1.1, XBB, and XBB.1.5) (Fig. 1c).

Memory B cells survive in the body for a long time after infection or vaccination, thus remaining ready to quickly recognize and combat any reinfection by pathogens[39,40]. We used flow cytometry to assess the dynamics of vaccine-mediated circulating memory B cells against the spike proteins from the WT strain and Delta and Omicron BA.1 variants (Supplementary Fig. 2). Generally, the percentages of memory B cells responding to the WT strain and Delta variant were increased after the second dose and further elevated with boosting (Fig. 1d, e). Notably, while the proportion of memory B cells recognizing the Omicron variant was unchanged after the second dose compared to the prevaccination level, it was strongly increased by the third booster immunization (Fig. 1f). This finding likely explains why the plasma neutralizing activity against the Omicron variant was nearly undetectable after the second vaccine dose but substantially increased after a third dose. Overall, we conclude that the humoral immunity elicited by the two-dose regimen may be less effective against pseudoviruses from the emerging SARS-CoV-2 variants; nonetheless, the three-dose boosting immunization with the inactivated vaccine that targets the original SARS-CoV-2 strain robustly elicits humoral responses in humans.

### The inactivated vaccine-induced B-cell responses

To examine the nature of the antibodies produced by memory B cells in response to the vaccination, we isolated mAbs by focusing on six individuals whose humoral immune responses were robust after the second vaccination (Supplementary Table 3). To determine whether there were changes in the antibodies produced by memory B cells after the booster vaccination, we obtained antibodies after Dose 3 from the same 6 individuals. In parallel, memory B cells of prevaccination samples from three participants in our cohort were sorted, and the V genes, complementarity-determining region 3 (CDR3) length and somatic hypermutation (SHM) were characterized by single-cell sequencing, from which paired heavy and light chains of IgG B-cell receptors (BCRs) ($n = 403$) represented the baseline circulating IgG memory cell repertoire and served as controls.

We obtained the heavy and light chains of 118 paired mAbs with binding activity from RBD-specific single B cells in the six individuals, who were sampled 1 month, 3 months and 6 months after the second vaccine dose and 1 month after the third dose (Supplementary Fig. 3

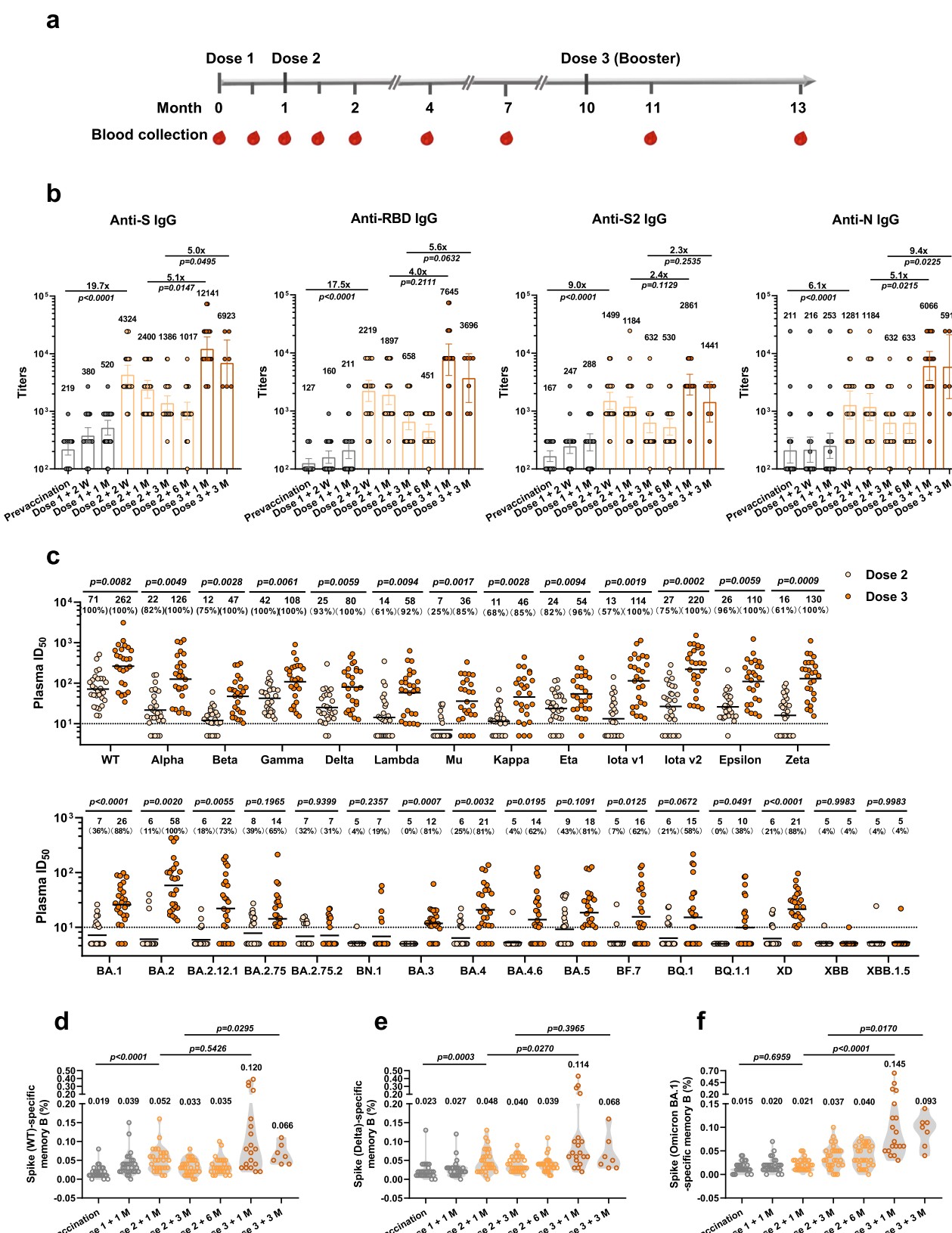

and Supplementary Table 3). Sixty-five of the 118 mAbs were isolated after the second vaccination, and 53 mAbs were obtained after the booster dose. Consistent with natural infection, IGHV3-30 and IGHV3-53, as well as IGKV1-39 and IGLV3-21, were significantly overrepresented in the RBD-binding memory B-cell compartment of vaccinated individuals (Fig. 2a and Supplementary Fig. 4). After the third

dose, the relative frequency of the IGHV3-9 and IGKV3-15 genes decreased, and unique clones were detected compared to those after the second dose (Fig. 2a and Supplementary Fig. 4). In addition, the average CDRH3 length was longer than that in healthy donors (Fig. 2b). The levels of SHM in the VH and VL genes of vaccinated individuals were significantly lower than those in healthy donors, and the

**Fig. 1 | SARS-CoV-2-specific humoral response in vaccinees. a** Schematic diagram of sample collection. Participants (*n* = 28) received a booster dose of BBIBP-CorV at a median of 9 months after two doses (4 weeks apart). Samples were collected before immunization (Prevaccination, *n* = 28); at 2 weeks (Dose 1 + 2 W, *n* = 28) and 1 month (Dose 1 + 1 M, *n* = 28) after the first dose; at 2 weeks (Dose 2 + 2 W, *n* = 28), 1 month (Dose 2 + 1 M, *n* = 28), 3 months (Dose 2 + 3 M, *n* = 28) and 6 months (Dose 2 + 6 M, *n* = 27) after the second dose; and at a median of 1 month (Dose 3 + 1 M, *n* = 19) and 3 months (Dose 3 + 1 M, *n* = 7) after the third dose (for details, see Supplementary Table 1). **b** Longitudinal plasma IgG titers to spike (S), receptor-binding-domain (RBD), the S2 region of Spike, and nucleocapsid (N) proteins measured by ELISA. Data are shown as geometric mean titers (GMTs) (values above bars) with 95% confidence intervals (CIs). The lower limit of quantification (LLOQ) was 100 for IgG titers. **c** Comparison of plasma neutralizing antibody titers to SARS-

CoV-2 pseudovirus in samples obtained 2 weeks after the second dose and 1 month and 3 months after the booster dose. Black bars and values above points represent the geometric mean 50% inhibitory dilution (ID$_{50}$), and the percentages of samples with detectable neutralizing activity above the LLOQ are given in parentheses. The horizontal dashed lines indicate the LLOQ. For values below the LLOQ, LLOD/2 values were plotted. The LLOQ of the assay for ID$_{50}$ was 10. Kinetics of SARS-CoV-2 memory B-cell responses to spike of wild type (WT) (**d**), Delta (**e**), and Omicron BA.1 (**f**) in vaccinated individuals. The numbers above the plots indicate mean values. (Also see Supplementary Fig. 2). Sample size in **b–f** is presented in (**a**). Statistical significance was determined using two-sided Kruskal–Wallis test with subsequent Dunn's multiple comparisons in (**b**) and (**d–f**), and two-sided multiple *t* test with subsequent Holm–Sidak method in (**c**).

mutation percentages increased after Dose 3 compared to those after Dose 2 (Fig. 2c). Furthermore, when mAbs from 1 month, 3 months, or 6 months after the second vaccination were stratified, an increase in SHM levels was observed over time for both the VH and VL genes (Supplementary Fig. 4c, d). Taken together, there was biased usage of IGHV, IGKV and IGLV genes in the vaccinated individuals, and a booster dose correlated with evolution of the memory B-cell compartment.

Subsequently, we exploited HIV-based SARS-CoV-2 S pseudo-typed viruses to measure the neutralizing activity of all 118 SARS-CoV-2-specific antibodies. Twenty antibodies that presented potent neutralizing activity against the SARS-CoV-2 WT strain were used to assess the neutralization spectrum against a panel of 28 SARS-CoV-2 variants (Supplementary Table 4). Eight of the 20 mAbs exhibited neutralizing activity against all the previously circulating VOCs (Alpha, Beta, Gamma, and Delta) and previously circulating VOIs (Lambda, Mu, Kappa, Eta, Iota v1, Iota v2, Epsilon, Zeta), and they were more susceptible to the Beta variant (Fig. 2d and Supplementary Fig. 5). Furthermore, eight mAbs were capable of inhibiting five Omicron subvariants (BA.1, BA.2, BA.2.12.1, BA.3, XD), and four of them retained activity against BA.4 and BA.5. However, the neutralizing activity of these mAbs was further impaired by the new Omicron subvariants with additional RBD mutations, including BA.2.75, BA.2.75.2, BN.1, BQ.1, BQ.1.1, XBB, and XBB.1.5. Overall, one of the 20 mAbs, 6-2C remained active against all the tested SARS-CoV-2 variants. Furthermore, we found that mAbs from samples collected after the second dose exhibited a greater loss of neutralizing potency against the pseudotyped variants that carry the E484K/A or N501Y mutation, while nearly half of the mAbs isolated after the third dose were more susceptible to variants with the L452R/Q mutation (Fig. 2d and Supplementary Table 5). Thus, neutralizing antibodies against various SARS-CoV-2 variants can be induced in individuals immunized with inactivated vaccines despite with compromised potency, and mAbs obtained after the primary two-dose regimen and the booster vaccination differ in their neutralization profile against the variants.

To determine the footprints of antibodies elicited by the inactivated vaccine, we analyzed the epitope specificity by 4 structurally defined mAbs, CB6 (Class 1), C121 (Class 2), COV2-2130 (Class 3), and COVA1-16 (Class 4), with biolayer interferometry (BLI)[41]. The competition profile indicated both overlapping and distinct epitopes recognized by these antibodies (Fig. 2e and Supplementary Fig. 6). We found that all but one of the seven antibodies isolated after the second dose blocked CB6 (Class 1), while nine of the 13 mAbs obtained after the booster dose did not compete with CB6 (Class 1) (Fig. 2e and Supplementary Fig. 6). Additionally, only two antibodies cloned after the second dose inhibited the binding of COV2-2130 (Class 3), while six of 13 mAbs obtained after the third dose competed with the Class 3 antibody (Fig. 2e and Supplementary Fig. 6). Thus, the inactivated vaccine could induce a broad spectrum of antibodies with distinct epitopes resembling those following natural infection, and

neutralizing antibodies isolated from samples collected after Dose 2 and Dose 3 had differential competition profiles and exhibited bias in epitope specificity. Notably, the broadly neutralizing antibody 6-2C was less competitive and did not compete with the potent antibodies 10-5B and 13-1C, indicating the recognition of distinct epitopes by the antibodies (Fig. 2e and Supplementary Fig. 7). Furthermore, all the evaluated antibodies blocked the binding of the RBD and hACE2 in the BLI assay (Supplementary Fig. 8). Finally, we measured the inhibition of infection of authentic SARS-CoV-2 by the nonoverlapping and potent antibodies 6-2C, 10-5B and 13-1C using a cytopathic effect (CPE) inhibition assay. Consistent with the data from the respective pseudovirus assay, the 6-2C and 10-5B antibodies were capable of neutralizing live SARS-CoV-2 wild type, Beta, Delta, Omicron BA.1 and Omicron BA.2, especially 10-5B, with IC$_{50}$ values below 150 ng/ml (Fig. 2f and Supplementary Fig. 9). 6-2C had moderate IC$_{50}$ values ranging from 11 to 1149 ng/ml. 13-1C was less potent against the Beta variant, although it could inhibit the other four authentic viruses (Fig. 2f and Supplementary Fig. 9). The potency of both antibodies was comparable to or greater than that of the approved antibody COV2-2130 as well as that of ADG-2 (Fig. 2f and Supplementary Fig. 9).

## Characterization of bispecific antibodies based on monoclonal antibodies

To further increase the neutralizing potency and breadth for resistance against viral evasion, one alternative is the use of multispecific antibodies, which have the advantages of both cocktails and single-molecule strategies. In this work, we designed and produced four bsAbs in the IgG-ScFv format based on the nonoverlapping and potent antibodies 6-2C/10-5B and 6-2C/13-1C (Fig. 2). The bsAbs BI-2C5B and BI-5B2C were the results of the combination of the mAbs 6-2C and 10-5B, with the position of the fragment antigen binding (Fab) of the mAbs exchanged with one another (Fig. 3a). Similarly, the bsAbs BI-2C1C and BI-1C2C were created from mAbs 6-2C and 13-1C (Fig. 3a). Both bsAbs folded correctly, and pure proteins could be obtained, as determined by sodium dodecyl sulfate-polyacrylamide gel electrophoresis (SDS-PAGE) (Supplementary Fig. 10).

The bsAbs bound with a low nanomolar affinity to the RBD of WT SARS-CoV-2 and to those of circulating variants of concern (VOCs), including Omicron BA.1, BA.2 and BA.4/BA.5 (Fig. 3b and Supplementary Fig. 11). The stronger affinity of bsAbs to the Omicron variant compared to that of their parental mAbs may be due to their multivalent binding to the RBD, which results in slower dissociation rates. The bsAbs also bound to the RBD that had already been saturated with either of the parental mAbs, which confirmed that both binding sites were functional (Fig. 3c, d). To assess the neutralizing activity of the bsAbs in vitro, we first determined the inhibition of SARS-CoV-2 infection with the panel of 28 pseudoviruses (Supplementary Table 2). The bsAbs neutralized WT SARS-CoV-2, previously circulating VOCs (Alpha, Beta, Gamma, Delta) and previously circulating VOIs (Lambda, Mu, Kappa, Eta, Iota v1, Iota v2, Epsilon, Zeta) at 1–18 ng/ml (IC$_{50}$), which was similar to or better than the activity of parental IgGs (Fig. 3e

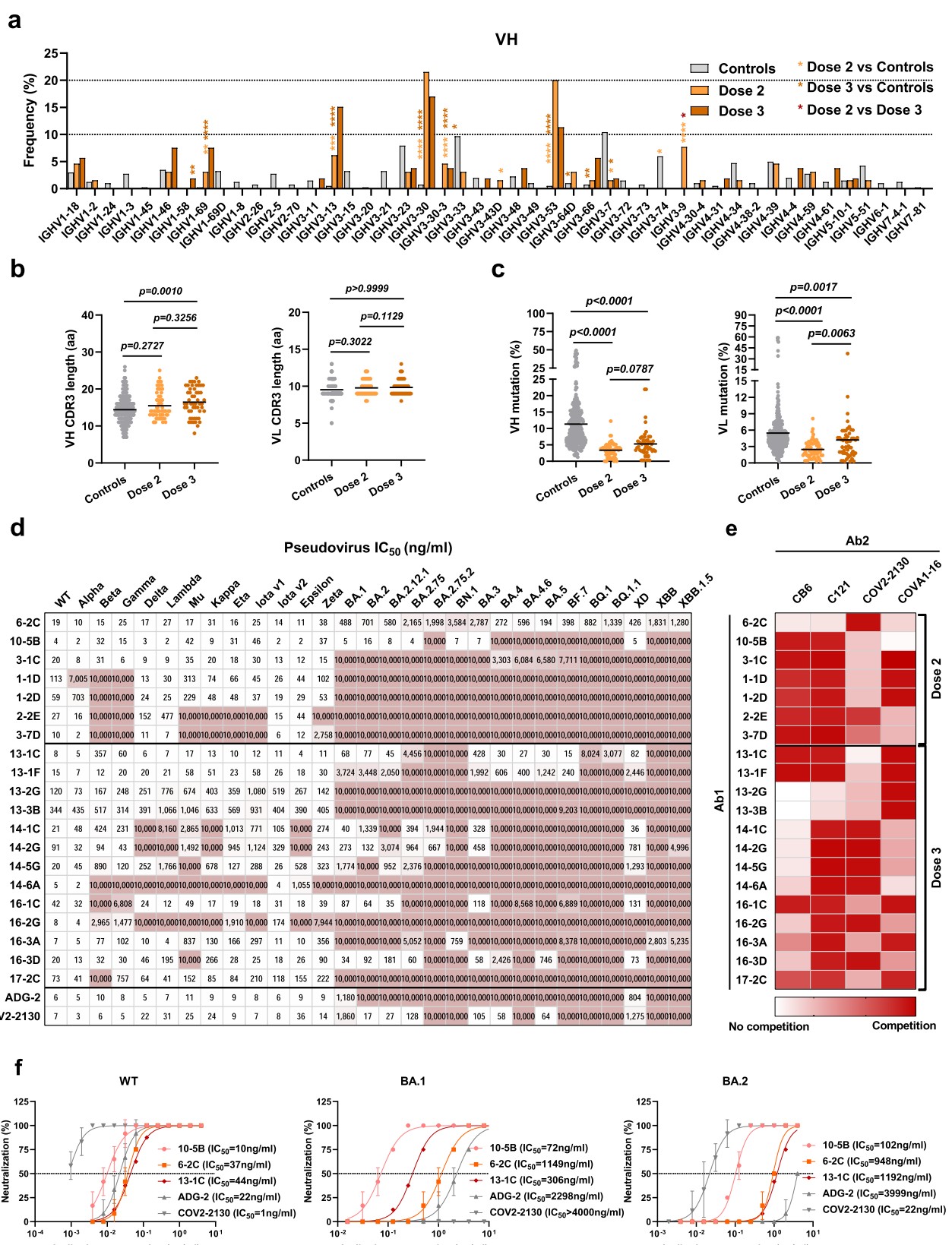

and Supplementary Fig. 12). The bsAbs remained effective against Omicron subvariants that bear escape mutations making them resistant to the individual mAbs (Fig. 3e). To confirm the activity of the bsAbs, we performed a cytopathic effect (CPE) inhibition assay with authentic viruses. All the bsAbs efficiently neutralized WT SARS-CoV-2 ($IC_{50}$ of 2–204 ng/ml), Beta ($IC_{50}$ of 34–220 ng/ml), Delta ($IC_{50}$ of

14–317 ng/ml), Omicron BA.1 ($IC_{50}$ of 3–58 ng/ml) and Omicron BA.2 ($IC_{50}$ of 15–306 ng/ml) (Fig. 3f). Nevertheless, BI-2C5B and BI-5B2C displayed stronger neutralization potency against the live viruses tested than BI-2C1C and BI-1C2C (Fig. 3f). We conclude that the in vitro binding and neutralization properties of the bsAbs make them preferable over their parental antibodies.

**Fig. 2 | Characterization of anti-SARS-CoV-2 RBD monoclonal antibodies from vaccinated individuals. a** The frequency distribution of human IGVH of anti-SARS-CoV-2 RBD mAbs (n = 118) from vaccinees compared to IgG-expressing memory B repertoires (n = 403) of healthy human donors. The mAbs were isolated from PBMCs sampled 1 month, 3 months and 6 months after the second vaccine dose (Dose 2) and 1 month after the third dose (Dose 3). Statistical significance was determined by two-sided Chi-square test with 1 degree of freedom (*p < 0.05, **p < 0.01, ***p < 0.001, ****p < 0.0001). **b** The amino acid (aa) length of the CDR3 at IGVH and IGVL in mAbs (n = 118) from vaccinated individuals and in those from prevaccinated healthy donors (n = 403). The horizontal bars indicate the mean values. Statistical significance was determined by two-sided Kruskal–Wallis test with subsequent Dunn's multiple comparisons. **c** The nucleotide somatic hypermutation levels of the V region in the heavy chain and light chain, as in (**b**). **d** Neutralization profile of pseudoviruses for 20 purified mAbs after Dose 2 and Dose 3, compared to those of two antibodies in clinical use or in development. The number in the box indicates half-maximal inhibitory concentration (IC$_{50}$) values. The color gradient indicates IC$_{50}$ values ranging from 0 (white) to 10,000 ng/ml. Antibodies with IC$_{50}$ values above 10,000 ng/ml were plotted as 10,000 ng/ml. Neutralization curves are shown in Supplementary Fig. 5. **e** Heatmap of the relative inhibition of the binding of the competing mAb (Ab2) to the preformed saturating mAb (Ab1)–RBD complexes (red, competition; light red, partial competition; white, no competition). BLI traces can be found in Supplementary Fig. 6. **f** Neutralizing activity of mAbs against live SARS-CoV-2 wild type, Omicron BA.1 and BA.2 for 10-5B, 6-2C and 13-1C, as well as COV2-2130 and ADG-2. The curves were fitted by nonlinear regression (log [inhibitor] vs. normalized response, variable slope). The dashed line indicates a 50% reduction in viral infectivity. Data for each mAb were obtained from a representative neutralization experiment. Mean ± s.d. of triplicates is shown, except for 13-1C (mean of duplicates). The experiment was replicated twice with similar results.

## Prophylactic activity of neutralizing antibodies against SARS-CoV-2 infection

To investigate the relationship between in vitro neutralization and protection in vivo against SARS-CoV-2, we first evaluated the prophylactic efficacy of 10-5B or 6-2C monotherapy or a combination of both 10-5B and 6-2C in a human ACE2 (hACE2) transgenic mouse model[42]. The mice were administered a single dose of 20 mg/kg 10-5B, 20 mg/kg 6-2C, or 10 mg/kg 10-5B with 10 mg/kg 6-2C one day before the viral challenge (Fig. 4a). The animals treated with PBS served as negative controls. In the challenge with the SARS-CoV-2 Delta variant, the mice treated with either individual antibody gained body weight more rapidly than the control mice during the first week of infection (Fig. 4b). The infectious viral levels in the lungs of antibody-treated animals were undetectable by the focus-forming unit (FFU) assay at 3 days post infection (dpi) (Fig. 4c). In the control group, the infected mice presented perivascular, peribronchial and alveolar inflammation, with the infiltration of immune cells and alveolar damage that are characteristic of viral pneumonia. In contrast, the mice treated with the antibodies showed notably less lung disease (Fig. 4d). Consistent with the histopathology results, the lungs of mice treated with PBS showed intense fluorescence at 3 dpi; however, there was minimal fluorescence in the 10-5B, 6-2C, and their combination-treated animals (Fig. 4e).

We also tested the bsAb BI-2C5B for prophylactic efficacy in a K18-hACE2 mouse model[43]. The mice received a single dose of 10 mg/kg BI-2C5B one day before intranasal challenge with a 3.6 × 10³ median tissue culture infectious dose (TCID$_{50}$) of the SARS-CoV-2 Omicron BA.2 variant (Fig. 4f). Passive transfer of BI-2C5B decreased Omicron BA.2-induced weight loss in the mice during the first week of infection compared to mice in the control group treated with PBS (Fig. 4g). Viral RNA levels in the lungs of the mice treated with BI-2C5B were reduced significantly at 3 dpi (Fig. 4h). Antibody-treated mice exhibited fewer lung lesions as well as reduced fluorescence intensity (Fig. 4i, j). Overall, these results indicated that 10-5B, 6-2C, and their combinations can effectively protect hACE2 transgenic mice against infectious SARS-CoV-2 Delta or Omicron BA.2 variants, suggesting their prophylactic potential for the management of the COVID-19 pandemic.

## Structural basis of antibody binding and neutralization

To define the molecular basis of mAb binding and neutralization and to understand how the activity of bsAbs might be increased, we investigated the mAbs 6-2C and 10-5B and bsAb BI-2C5B for structural characterization by X-ray crystallography and cryoelectron microscopy (cryo-EM). These structures included a crystal structure of WT-RBD bound with 6-2C Fab (2.2 Å) and cryo-EM structures of WT spike bound with 10-5B Fab (3.3 Å), BA.1 spike bound with 10-5B and 6-2C Fabs (3.2 Å), and BA.4 spike bound with 10-5B and 6-2C (2.9 Å) Fabs (Supplementary Figs. 13–16, 17a and Supplementary Tables 6 and 7). We further performed local refinement to improve the densities around the antibody/RBD interfaces in the cryo-EM structures (Supplementary Fig. 16). Structural alignments showed that 6-2C or 10-5B had nearly identical binding modes to WT, BA.1 and BA.4 spike proteins (Supplementary Fig. 17b). Therefore, we utilized the cryo-EM structure of the BA.1 spike bound with 6-2C and 10-5B Fab to illustrate the overall features of antibody recognition. The 3D classification showed a major conformation of the BA.1 S trimer with three RBDs adopting the "up" conformation, each bound by one 6-2C Fab and one 10-5B Fab (Fig. 5a). The neutralizing epitope on the RBD has been grouped into four Classes (Class 1–Clas 4) or seven "communities" (RBD-1 to RBD-7)[41,44]. 6-2C binds to the core with an uncommon epitope between RBD-5 and RBD-7 (Class 3), consisting of residues N343, A344, T345, L371, P373, F374, F375, W436, N437, S438, N439, K440, L441, S443, K444, V445, P499, T500, V503, Q506, Y508, and N343-linked glycans, which has slight overlap with the ACE2-binding site (Fig. 5b, c and Supplementary Fig. 17c). 10-5B binds to the residues Y449, L455, F456, N477, V483, A484, G485, F486, N487, C488, Y489, and R493 on the receptor-binding motif (RBM). This epitope belongs to the RBD-2 community (Class 1) and has extensive overlap with the ACE2-binding site on the RBM (Fig. 5b, c and Supplementary Fig. 17c).

Notably, 6-2C broadly neutralized all the tested variants, including the most antibody-evasive variants XBB.1.5, XBB, BQ.1.1, and BA.2.75.2. In general, core-RBD epitopes tend to be mutationally constrained with respect to folding and expression and are mostly conserved across sarbecoviruses, explaining the possible neutralization breadth of antibody 6-2C[45]. Nevertheless, its potency against the tested Omicron subvariants was reduced by 10- to 189-fold compared to its potency against WT SARS-CoV-2 (Fig. 2d). The interface of 6-2C with WT RBD was well defined in the 2.2 Å crystal structure, and densities around its interfaces with BA.1 and BA.4 RBDs in the cryo-EM structures were also greatly improved by local refinement for model building of the side chains of most interacting residues. PISA was used to calculate the interface areas, contacts, and hydrogen bonds at the interfaces of 6-2C with WT, BA.1 and BA.4 RBDs, respectively. The number of contacts at the 6-2C/WT RBD and 6-2C/BA.4 RBD was nearly the same (145 vs. 141) and increased slightly to 167 at the 6-2C/BA.1 RBD-interface (Fig. 5d). The number of hydrogen bonds was reduced from 16 for WT RBD to 9 for BA.1 RBD and slightly to 13 for BA.4 RBD (Fig. 5d). Structural analysis showed that the disruption of hydrogen bonds was mainly due to the S371L/F, S373P, and S375F mutations found in all Omicron subvariants, resulting in a main-chain conformational change of the hairpin loop (residues Y369–C379) harboring these three mutations. For example, 5 hydrogen bonds from WT-RBD S371, A373 and S373 to 6-2C were all disrupted at the 6-2C/BA.1-RBD interface due to conformational change. Only 2 hydrogen bonds formed between BA.4-RBD F371 and 6-2C (Fig. 5e). The distinct loops of BA.1 (aa371–aa377, LAPFFTF) and BA.4 (aa371–aa377, FAPF-FAF) and the resulting side-chain conformational differences led to the different contacts for antibody 6-2C with the spike of BA.1 and BA.4. (Supplementary Fig. 18). These observations together explained the

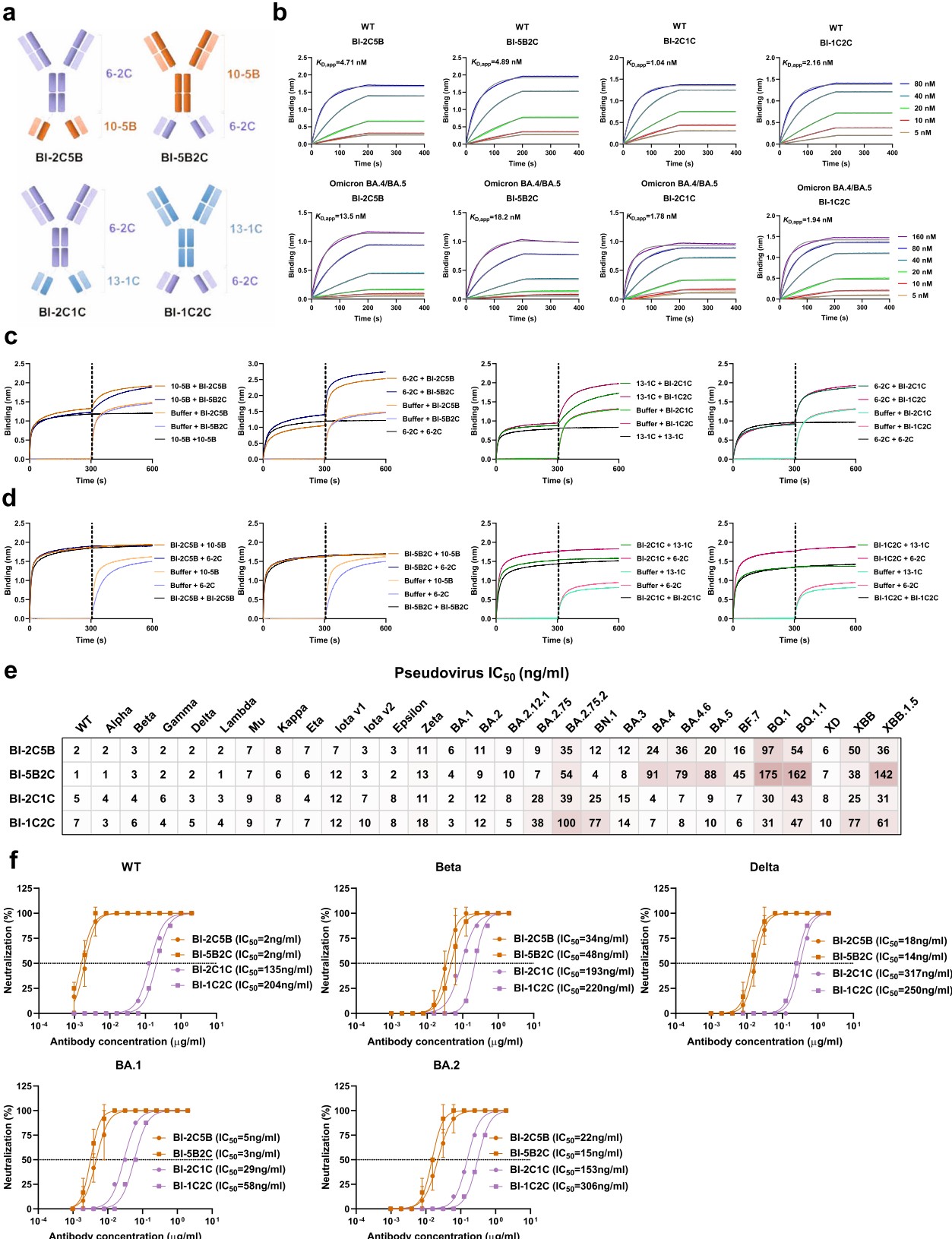

relatively decreased binding activity of 6-2C against the Omicron variant. Furthermore, the epitope of 6-2C overlapped or clashed with the ACE2-binding site on the SARS-CoV-2 RBD, indicating competition with ACE2, as supported by the BLI assay (Fig. 5c and Supplementary Fig. 19), suggesting that 6-2C neutralized SARS-CoV-2 through the inhibition of virus/host cell interactions.

We have shown that 10-5B is highly potent against the pseudoviruses of WT SARS-CoV-2 and nearly all tested VOCs and VOIs, including Omicron BA.1, BA.2, BA.2.12.1, BA.3 and XD subvariants, with IC$_{50}$ values in the range of 2–46 ng/ml (Fig. 2d). However, its potency was significantly decreased to more than 10,000 ng/ml against Omicron BA.4/5 and its subvariants (Fig. 2d). Similar to the analysis of the

**Fig. 3 | Design and characterization of bispecific antibodies (bsAbs).**
**a** Schematic of four bispecific antibodies in IgG-ScFv format. The parental mono-clonal antibodies that form the bispecific antibodies are color-coded: purple, 6-2C; orange, 10-5B; blue, 13-1C. **b** Binding affinity of bsAbs to RBDs of WT and Omicron BA.4/BA.5. Biotinylated SARS-CoV-2 RBDs were loaded onto the surface of SA biosensors. Individual antibodies were at a series of concentrations. The associa-tion and dissociation of response curves of the antibodies are shown. The gray lines represent the fitted curves based on the experimental data. Apparent dissociation constants ($K_{D,app}$) are shown above each plot. **c, d** Competitive binding of parental monoclonal antibodies and bispecific antibodies. In **c**, the immobilized RBD com-plexed with parental monoclonal antibodies (first antibody) binds to bispecific antibodies (second antibody). In **d**, RBD–bsAbs prevent binding by their parental monoclonal antibodies. **e** Neutralization profile of SARS-CoV-2 pseudoviruses for the two bispecific antibodies. The number in the box indicates $IC_{50}$ values. The color gradient indicates $IC_{50}$ values ranging from 0 (white) to 10,000 ng/ml. The experiments were performed at least twice. Neutralization curves are shown in Supplementary Fig. 12. **f** Neutralizing activity of bispecific antibodies against live SARS-CoV-2 wild type, Beta, Delta, Omicron BA.1, and Omicron BA.2 variants. Authentic SARS-CoV-2 neutralization was performed using a cytopathic effect (CPE) assay. The curves were fitted by nonlinear regression (log [inhibitor] vs. normalized response, variable slope). The dashed line indicates a 50% reduction in viral infectivity. Data for each bsAb were obtained from a representative neu-tralization experiment. Mean ± s.d. of triplicates (BI-2C5B, BI-5B2C) or mean of duplicates (BI-2C1C, BI-1C2C) is shown. The experiment was replicated twice with similar results.

6-2C/RBD interfaces, we used PISA to calculate the interface areas, contacts, and hydrogen bonds at the interfaces of 10-5B with WT, BA.1 and BA.4 RBDs, respectively (Fig. 5d). Although the interface density map of 10-5B was not as good as that of 6-2C in the locations of the side chains of interacting residues, the PISA analysis still provided rational information for investigating the structural basis of the binding and neutralization of 10-5B against Omicron subvariants. It was found that there were ~86 contacts and ~10 hydrogen bonds at the 10-5B/WT-RBD interface (Fig. 5d). At the 10-5B/BA.1-RBD interface, the number of contacts was ~84, but the number of hydrogen bonds was decreased to ~6. The 10-5B/BA.4-RBD interface showed more significant reductions: the number of contacts was decreased to ~65 and the number of hydrogen bonds was only ~1 (Fig. 5d). Structural analysis further revealed that most hydrogen bonds formed between the WT-RBD E484–Y489 region and 10-5B. The S477N and E484A mutations in BA.1 and BA.4 affected the formation of hydrogen bonds of these two RBD sites with 10-5B (Fig. 5f). The additional mutation F486V found in BA.4 further disrupted more hydrogen bonds by changing the local struc-ture and breaking the hydrogen bond network around the E484–Y489 region. These findings explained the relative reduction in the binding affinity of 10-5B with BA.1 RBD, and the strikingly decreased binding with BA.4 RBD. The footprint of 10-5B largely overlapped with the site of ACE2 in the RBM, resulting in competition with ACE2 binding to the RBD via steric hindrance, providing the structural basis for its neu-tralization of SARS-CoV-2 WT and Omicron BA.1 (Fig. 5c and Supple-mentary Fig. 19).

The bsAb BI-2C5B designed based on 6-2C and 10-5B potently and broadly neutralized all the tested variants with better $IC_{50}$ values than its parental antibodies. To understand the mechanism of the increased neutralization, we tried to determine the cryo-EM structure of BI-2C5B bound to the SARS-CoV-2 S trimer. However, large aggregates were observed in the cryo-EM images of BA.4 S trimer-bsAbs BI-2C5B com-plex that were not observed in the cryo-EM images of the BA.4 S trimer-10-5B-6-2C ternary complex (Supplementary Fig. 20). This finding indicated that the increased activity of the bsAbs might be associated with avidity effects through which the bsAb crosslinked adjacent spike proteins to achieve tighter binding. Overall, using combinations of mAbs with cooperative function or bsAbs would both increase potency and decrease the risk of escape. Nonetheless, the polyclonal nature of the antibody response and the diversity of epitope specificity elicited by inactivated vaccination would likely help limit full vaccine escape when encountering the emerging variants.

## Discussion

Inactivated COVID-19 vaccines are being globally exploited to prevent SARS-CoV-2 infection. Here, we provide data offering molecular insights into the longitudinal immune responses to an inactivated vaccine, BBIBP-CorV, in 28 participants who received a booster vacci-nation after the primary two-dose regimen. In the cohort of vaccinees, we first assessed the humoral responses to SARS-CoV-2 after BBIBP-CorV immunization. Longitudinal analysis suggested that a third dose

of the inactivated vaccine was necessary to induce a marked immune response, which is consistent with the results for other SARS-CoV-2 vaccines[46–49]. The booster dose significantly increased the binding and neutralizing antibody titers that waned several months after the sec-ond dose[50,51]. Moreover, neutralization titers against the Omicron subvariants were induced in the majority of the recipients after the third dose of vaccine despite being strikingly impaired by the new variants with additional mutations in the RBD. Notably, memory B cells specific for the spike protein were detected in most of the vaccine recipients with no significant decrease in levels at 6 months after the second dose. Our results also showed a measurable memory B-cell response to the Delta and original Omicron variants. A third dose of BBIBP-CorV efficiently elicited recruitment and expansion of these memory B cells, resulting in amplification of antibody responses cap-able of neutralizing SARS-CoV-2 variants, including Omicron sublineages[52]. The boosted antibody responses subsequently declined after ~3 months but remained significantly above the levels observed at the same timepoint after the second dose.

We found that four individuals showed higher percentages of spike-specific memory B cells at one month after the third vaccination. The four individuals also exhibited high anti-spike/RBD IgG titers and high neutralization titers after the third vaccination (Supplementary Table 8). However, there was no significant difference in the factors (age, sex, BMI and time since vaccination) associated with high antibody levels between the four individuals and other participants[53,54]. Addi-tionally, their plasma activity and percentages of spike-specific memory B cells before the third vaccination showed distinct levels. Furthermore, it was difficult to analyze the samples at the molecular level since only one was among the six individuals for mAb isolation. Thus, the higher percentages of spike-specific memory B cells in these individuals may be attributed to the inter- and intra-person heterogeneity as the circu-lating B-cell populations of individuals are highly individualized and extremely diverse[55]. Additionally, antibody responses varied sub-stantially in convalescent and vaccinated individuals[56–58]. Similarly, there was extensive person-to-person variation in how mutations affect plasma antibody binding and neutralization[59].

When assessing RBD-binding memory B cells, IGHV3-30, IGHV3-53, IGKV1-39, and IGLV3-21 were highly overrepresented in vaccinated individuals, which is consistent with the findings in COVID-19 con-valescents and those immunized with mRNA vaccines[26,60]. Among S trimer-specific antibodies produced during natural infection, the IGHV3-30, IGKV3-20, and IGHJ6 genes were enriched, while the fre-quencies of IGHV3-53, IGKV1-39, and IGLV3-21 were not significantly increased[61], suggesting that the latter might be genetic features of RBD-specific antibodies. The characteristics of S trimer-specific antibodies produced in response to inactivated vaccines remain to be determined. Additionally, we found that RBD-binding memory B cells had low SHM levels in VH and VL genes, in line with the near-germline IgG antibodies against SARS-CoV-2 in natural infection, suggesting that these cells were primed by the ongoing vaccination and differentiated from B cells without extensive germinal center experience[30,61–63]. The repertoire of

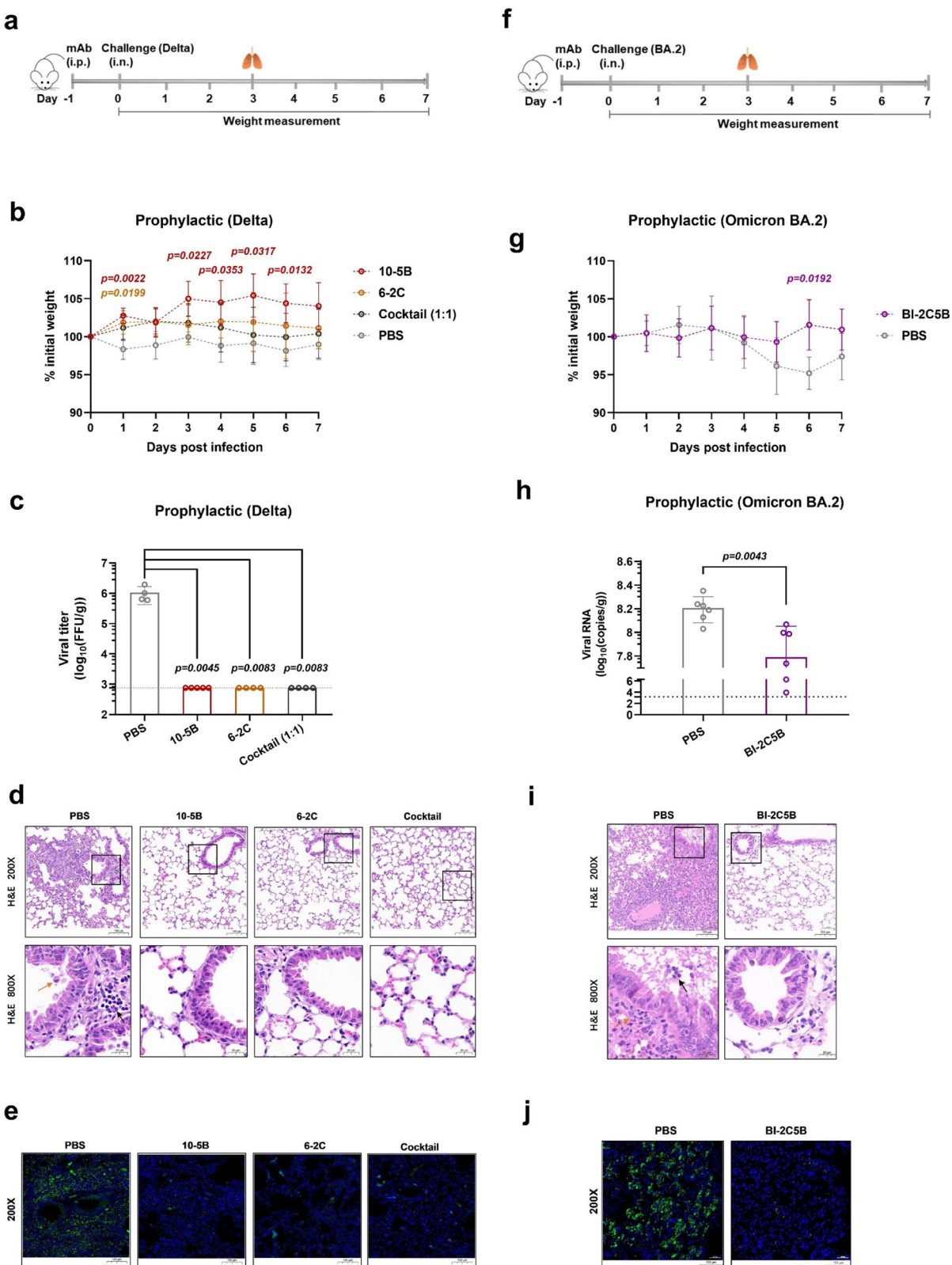

memory B-cell receptors from controls encodes a comprehensive record of an individual's immunological encounters, usually experiencing continued development, and so have higher SHM levels than those of RBD-binding memory B cells[55]. Notably, the timing of mAb isolation following vaccination should be considered, as the SHM level increased over time in both our work and previous reports[26,64], indicating continued development of the humoral response.

We compared the neutralizing mAbs obtained after the second dose and the booster dose. The average breadth of mAbs from dose 3 (54.4%) was slightly higher than that of those from dose 2 (49.8%) when we calculated the percentage of pseudoviruses ($n = 29$) to which the mAbs had an $IC_{50}$ value below 10,000 ng/ml. However, the number of neutralizing mAbs was relatively small and might not be sufficient to analyze the difference. Another possible reason is that the mAbs were

**Fig. 4 | Prophylactic activity of antibodies in humanized ACE2 mouse models.**
**a** Female 9- to 12-week-old hACE2 transgenic mice were received one dose (20 mg/kg body weight) of 10-5B, 6-2C, or their combination (1:1 ratio) 24 h prior to intranasal challenge with $5 \times 10^4$ FFU of SARS-CoV-2 Delta variant. **b** Body weight change was recorded over the course of 7 days ($n = 5$). Data are shown as the mean ± s.d. of each experimental group. The dark red $P$ value represents a comparison between 10-5B and the control, and the dark yellow $P$ value represents a comparison between 6-2C and the control. **c** The viral burden from mouse lungs at 3 days post infection (dpi) was determined by a focus-forming unit (FFU) assay (10-5B, $n = 5$; other groups, $n = 4$). **d** Representative hematoxylin and eosin staining of lung sections from infected mice (3 dpi). Bronchial epithelial cell desquamation (yellow arrow) and lymphocyte infiltration (black arrow) are shown. Scale bars: 100 μm (top); 20 μm (bottom). **e** Representative immunofluorescence staining of lung sections from infected mice (3 dpi). The nucleocapsid (green) was stained with Alexa Fluor 488-conjugated anti-rabbit IgG. The nuclei (blue) were stained with DAPI. Scale bars: 100 μm. **f** Female 7- to 8-week-old K18-hACE2 mice received one dose (10 mg/kg body weight) of bsAb BI-2C5B 24 h prior to intranasal challenge with $3.6 \times 10^3$ TCID$_{50}$ of SARS-CoV-2 Omicron BA.2 variant. **g** Body weight change was recorded ($n = 6$) as in (**b**). The purple $P$ value represents a comparison between BI-2C5B and the control. **h** The viral load from mouse lungs (3 dpi) was measured by RT-qPCR assay ($n = 6$). **i, j** Representative H&E and immunofluorescence staining of lung sections (3 dpi) from infected K18-hACE2 mice, as in (**d**) and (**e**), respectively. $P$ values were determined using two-way analysis of variance (ANOVA) with Tukey's post hoc test in (**b**), two-sided Kruskal–Wallis test with subsequent Dunn's multiple comparisons in (**c**), two-sided multiple t test with subsequent Holm–Sidak method in (**g**), and two-tailed Mann–Whitney $U$ tests in (**h**). The dashed line indicates the assay limit of detection, and the mean ± s.e.m. of all data points are shown in (**c**, **h**).

isolated from a subset ($n = 6$) of participants. Therefore, bias may exist because not all samples were equally evaluated. If Omicron RBD is used for B-cell isolation, we expect mAbs to exhibit enhanced breadth. WT vaccine-elicited antibodies that bind and neutralize Omicron probably target conserved sites shared with the WT strain[52,65]. Additionally, when screening for binding activity with the transfection supernatants, we found that Omicron RBD-binding mAbs simultaneously bound the WT, Alpha, Beta, and Delta strains. Nevertheless, studies are warranted to further investigate this possibility.

The distinct neutralization profiles between mAbs isolated after the second vaccine dose and the booster dose broadly corresponded to the difference in the competitive pattern of these mAbs. Four of seven mAbs (1-1D, 1-2D, 2-2E, 3-7D) obtained after Dose 2 presented impaired neutralizing activity against variants carrying the N501Y and/or E484K/A mutation, which could be explained by their competitive binding with Class 2 antibody (C121). This finding was consistent with prior reports showing that the E484K mutation was associated with resistance to class 2 mAbs[66,67]. Although 1-1D and 1-2D displayed better tolerance to single E484K/A mutation, it rendered them inactive when combined with N501Y. Unlike the neutralizing mAbs isolated after Dose 2, nearly half of the neutralizing mAbs isolated after Dose 3 were more susceptible to variants with the L452R/Q mutation (Delta, Lambda, Kappa, Epsilon, et al), and epitope analysis revealed that all these antibodies competed with the Class 3 antibody (COV2-2130), supporting previous findings showing that L452 substitutions escaped Class 3 antibodies[11]. Additionally, as revealed by deep mutational scanning, mutations at F486 also had substantial antigenic effects on antibodies, which rendered the candidate antibody 10-5B inactive due to breaking the hydrogen bond network around the E484–Y489 region[59]. Furthermore, another promising antibody, 13-1C, was extensively affected by the new Omicron subvariants with the convergent mutation N460K in the RBD[68]. Finally, the potency of the broadly neutralizing mAb 6-2C was impaired by the additional mutations R346T and K444T in the RBD, similar to the neutralization features of Class 3 mAbs[69].

Despite the differential competition profiles of neutralizing antibodies isolated after Dose 2 and Dose 3, no significant difference was observed in competitive antibody titers on these four classes in plasma (Supplementary Fig. 21). Additionally, we analyzed the six samples from which mAbs were isolated, to exclude the possible influence of selection bias. However, no skews in epitope specificity were observed from the plasma samples. The discrepancy in competition profiles between neutralizing mAbs and plasma might be due to the impact of nonneutralizing antibodies in the plasma. Nonneutralizing antibodies comprised approximately 20–45% of the RBD-specific antibodies and displayed distinct competition features from the neutralizing antibodies in natural infection[64]. In our work, nonneutralizing antibodies accounted for ~80% of the isolated mAbs and might exert their influence on the competitive results. In addition, this lack of concordance between the epitopes of plasma and mAbs is consistent with reports of other studies showing that the specificities of potent mAbs often do not recapitulate the plasma from which they were isolated[59,66,70].

Furthermore, we found that all nine IGHV3-53/IGHV3-66-derived antibodies of 20 neutralizing mAbs competed with Class 1 and Class 2 RBD-targeting antibodies, which was consistent with the shared footprint on RBD and the similar angle approaching the RBD compared to that of ACE2[41,70,71]. In addition, IGHV3-30 and IGHV1-69 gene-encoded neutralizing mAbs were classified into Class 2 and Class 3 of RBD-targeting antibodies. Similarly, IGHV1-58 was enriched in mAbs isolated from Dose 3, and IGHV1-58 gene-encoded neutralizing tended to bind the left shoulder of RBD, often focusing on the far tip[72]. Nevertheless, no obvious relationship between VL gene preference and structural features was observed in our study, possibly due to the relatively low contribution of the light chain to epitope binding.

Approved antibodies and our data confirmed that it was difficult for monotherapy to persistently resist the continuously emerging mutants of SARS-CoV-2[68,69,73,74]. BsAbs targeting nonoverlapping epitopes are preferred for increased efficacy and improved resistance against viral escape[75–77]. In this work, we designed four bsAbs on the basis of nonoverlapping monoclonal antibodies derived from donors who had received the BBIBP-CorV inactivated vaccine. The bsAbs had greatly enhanced binding affinity to the RBD of WT SARS-CoV-2 and its VOCs and could neutralize variants that evaded the parental mAbs. When we investigated neutralization mechanism via structure determination, large aggregates were observed in S trimer-bsAb complexes. This finding indicated that the increased activity of bsAbs might be due to their higher potential for inter-spike crosslinking[30,78].

The persistence of COVID-19 has led to the continuous generation of mutational variants, which largely circumvent the immune barriers in human populations fortified by vaccination and natural infection[79–81]. The SARS-CoV-2 Omicron subvariants have been reported to exhibit resistance to neutralization by all available therapeutic antibodies and sera from vaccinees, posing a high reinfection risk in humans[68,69,82]. Furthermore, the dominant spread of the immune-evasive Omicron sublineages may lead to much higher burdens on global public health[83]. Therefore, antibodies with more broadly neutralizing activity, which could prevent infection by known and future variants, represent an alternative strategy for both therapeutic and prophylactic interventions during the COVID-19 pandemic. In addition, these antibodies can also be used to identify highly conserved antigenic determinants across various coronavirus strains to guide the design of a broad-spectrum vaccine and to serve as indicators for cross-protection potential upon vaccine immunization. In this study, we found that the representative mAb 6-2C elicited by an inactivated vaccine broadly neutralized all the tested SARS-CoV-2 variants. The bsAbs based on 6-2C and 10-5B further increase the potency and breadth of neutralization against viral evasion in vitro and in vivo. Overall, although the effect of COVID-19 vaccines for preventing infection is generally diminished now, the broad spectrum of neutralizing antibodies induced in individuals who are immunized with

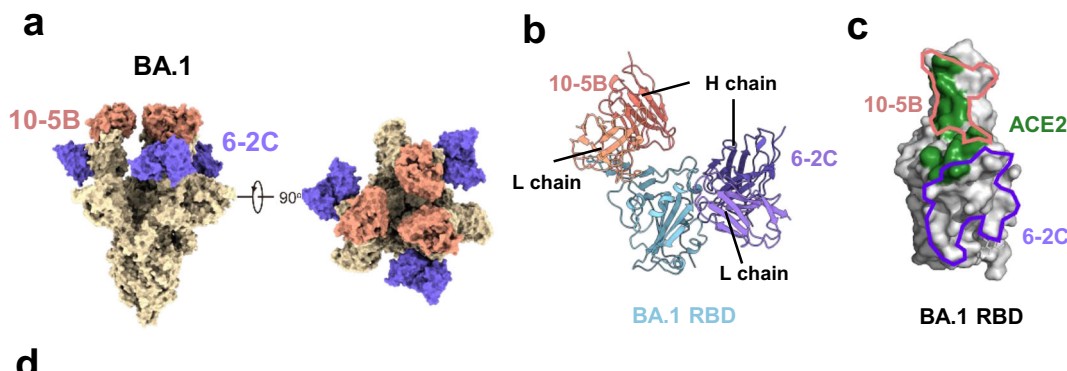

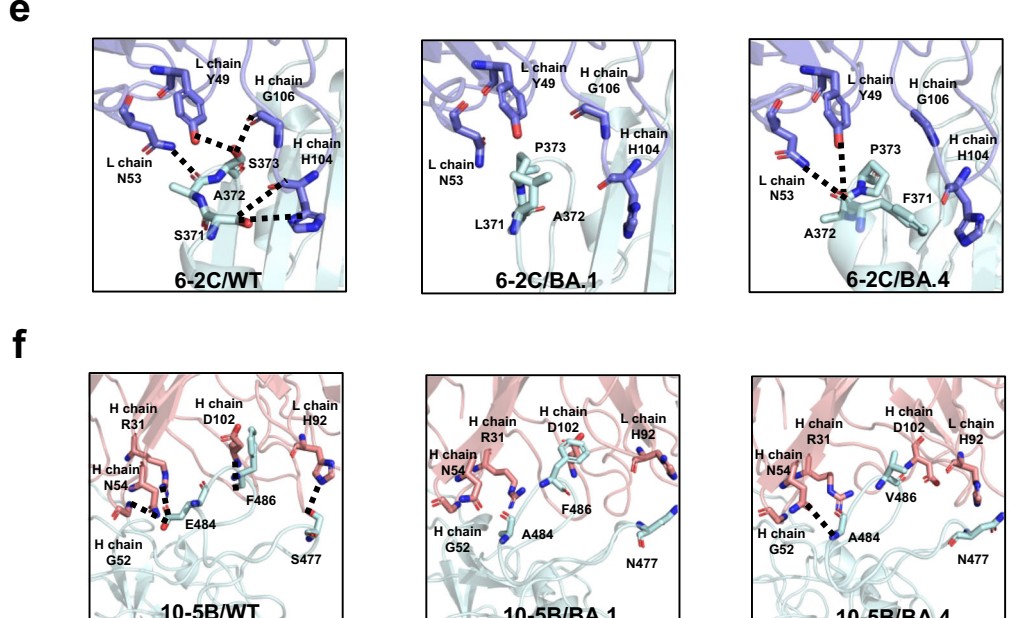

| | | 6-2C | | | 10-5B* | | |
|---|---|---|---|---|---|---|---|
| | | **H chain** | **L chain** | **Total** | **H chain** | **L chain** | **Total** |
| **Interface (Å²)** | WT | 660 | 412 | 1072 | 534 | 130 | 664 |
| | BA.1 | 705 | 488 | 1193 | 530 | 197 | 727 |
| | BA.4 | 694 | 419 | 1113 | 473 | 175 | 648 |
| **Contacts** | WT | 93 | 52 | 145 | 70 | 16 | 86 |
| | BA.1 | 82 | 85 | 167 | 65 | 19 | 84 |
| | BA.4 | 92 | 49 | 141 | 53 | 12 | 65 |
| **H bonds** | WT | 9 | 7 | 16 | 9 | 1 | 10 |
| | BA.1 | 3 | 6 | 9 | 4 | 2 | 6 |
| | BA.4 | 10 | 3 | 13 | 1 | 0 | 1 |

**Fig. 5 | Structural characterization of the mAbs 6-2C and 10-5B. a** Cryo-EM structure viewed along two orthogonal orientations of the SARS-CoV-2 Omicron BA.1 S trimer (yellow) bound with 10-5B Fab (salmon) and 6-2C Fab (purple). The three "up" RBDs are bound by three 10-5B Fabs and three 6-2C Fabs, respectively. **b** Cartoon diagram of one 10-5B Fab (heavy chain in deep salmon and light chain in salmon) and one 6-2C Fab (heavy chain in deep purple and light chain in light purple) bound to one SARS-CoV-2 RBD (cyan). **c** The superposition of 10-5B (salmon), 6-2C (purple) and ACE2 (green) footprints on the BA.1 spike RBD. **d** Table of 6-2C/10-5B and RBD (WT, BA.1, and BA.4) interactions, including interface area, contacts, and hydrogen bonds. *The values are approximations. **e** H bonds of 6-2C formed with RBD (WT, BA.1, and BA.4) residues 371, 372, and 373. H bonds are marked by dotted lines. **f** H bonds and salt bridges of 10-5B formed with RBD (WT, BA.1, and BA.4) residues 477, 484, and 486.

inactivated SARS-CoV-2 vaccines is probably effective at reducing the risk of hospitalization and severe disease as well as possibly postacute sequelae, resulting in potential application for protection against future COVID-19 epidemics.

## Methods
### Study design
We recruited individuals who were planning to immunized with COVID-19 vaccines in May 2021. Twenty-eight healthy volunteers were enrolled

for blood donation and followed regularly to evaluate the immune response to approved COVID-19 vaccines. Vaccination was conducted as part of routine care outside of the cohort study. Participants received the primary two-dose regimen (4 weeks apart) of the BBIBP-CorV (or Covilo) vaccine and were boosted at a median of 9 months after the second dose with the same vaccine based on information provided by the participants during regular follow-up visits. BBIBP-CorV, an inactivated COVID-19 vaccine, was developed by Sinopharm's Beijing Institute of Biological Products (China) and became the first whole inactivated virus COVID-19 vaccine to receive emergency use authorization by the WHO. The HB02 strain with optimal replication and virus yields was selected and passaged in Vero cells to generate vaccine production. The BBIBP-CorV stock was inactivated by thoroughly mixing with β-propionolactone at a ratio of 1:4000 at 2–8 °C. The vaccine was manufactured as a liquid formulation containing 4 µg total protein with aluminum hydroxide adjuvant (0.45 mg/ml) per 0.5 ml[84]. Eligible participants were healthy people 18 to 65 years of age, and key exclusion criteria included contraindications for vaccines, previous diagnosis of COVID-19, previous vaccination with any coronavirus vaccine, known acute and chronic infectious diseases, severe chronic and bleeding disorders, pregnancy, and lactation. Blood of the participants was collected before and after immunization (for details see Supplementary Table 1). The work was approved by the Institutional Review Board of Tsinghua University (20210061). Written informed consent was obtained from all participants.

## Blood sample processing and storage

Blood samples were collected from participants at the study visit and processed within 12 h. Briefly, the plasma and blood cells were separated by centrifugation, and the blood cells were subjected to Ficoll Paque Plus (GE Healthcare, 17144002) after a 1:1 dilution in PBS (Gibco, C10010500BT) to isolate peripheral blood mononuclear cells (PBMCs) according to the manufacturer's instructions. The plasma was divided into aliquots and stored at -80 °C. PBMCs were stored in liquid nitrogen in the presence of 10% dimethyl sulfoxide (DMSO) (Sigma, D2650-100 ml) in fetal bovine serum (FBS) (HyClone, SH30084.03). Before experiments were performed, aliquots of the plasma samples were heat-inactivated (56 °C for 1 h) and then stored at 4 °C.

## Enzyme-linked immunosorbent assay (ELISA)

The SARS-CoV-2 S trimer, RBD, S2 and nucleocapsid protein (N) (Acro Biosystems SPN-C52H9, SPD-C52H3, S2N-C52H5, and NUN-C5227, respectively) were coated on 96-well plates at 0.5 µg/ml in PBS overnight at 4 °C. After the plates were blocked with buffer (1× PBS with 3% BSA (Solarbio, A8020) and 0.05% Tween-20 (Sigma, P9416)) for 1 h at 37 °C, plasma samples were added and incubated for 1 h at 37 °C. The plasma samples were assayed at a 1:100 starting dilution and 7 additional threefold serial dilutions in blocking buffer (1× PBS with 1% BSA (Solarbio, A8020) and 0.05% Tween-20). The plates were washed 5 times with washing buffer and then incubated with anti-human IgG (Abcam, ab97225) or IgM (Abcam, ab97205) secondary antibody conjugated to horseradish peroxidase (HRP) in blocking buffer at a 1:50,000 dilution (and IgG) or 1:20,000 dilution (IgM) for 0.5 h at 37 °C. After the plates were washed 5 times, the HRP substrate TMB (Solarbio, PR1200) was added for 10 min, followed by the addition of 50 µl of 1 M $H_2SO_4$ (Solarbio, C1058) to stop the reaction. The absorbance was measured at 450 nm with an ELISA microplate reader (TECAN Infinite 200 PRO).

## SARS-CoV-2 pseudotype neutralization assays

SARS-CoV-2 and variant pseudotypes were generated by cotransfection of the S-glycoprotein-encoding plasmid and human immunodeficiency virus (HIV) backbone expressing firefly luciferase (pNL4-3.luc.RE) into HEK293T cells (ATCC, CRL3216) using Lipofectamine 2000 transfection reagent (Life Technologies, 11668-019) according to

the manufacturer's instructions[85]. Pseudotyped virus stocks were collected 48 h after transfection, filtered and stored at −80 °C. Viral titers were measured based on luciferase activity determined by relative light units (Luciferase Assay Systems, Promega Biosciences). Plasma samples or antibodies were serially diluted threefold and then incubated with SARS-CoV-2 pseudovirus for 1 h at 37 °C. HeLa-hACE2 cells (Prof. Qiang Ding, Tsinghua University; HeLa, ATCC CCL-2) ($1.3 × 10^4$ per well) were directly added to the antibody-virus mixture. After 48 h, the cells were washed with PBS and lysed with Luciferase Cell Culture Lysis 5× reagent (Promega, E1531), and the luciferase activity was measured using the Luciferase Assay System (Promega, E1501). Plasma or mAbs were tested in duplicate wells, and the assay was independently repeated at least twice for sets of plasma samples and individual mAbs. The neutralization $ID_{50}$ or $IC_{50}$ was calculated using nonlinear regression (log [inhibitor] vs. normalized response, variable slope) (GraphPad Prism v.8.0).

## Authentic SARS-CoV-2 neutralization assays

An authentic SARS-CoV-2 neutralization assay was performed using a cytopathic effect (CPE) assay in a biosafety level 3 laboratory. Briefly, each antibody was serially diluted twofold starting at 2 µg/ml (Beta), 4 µg/ml (WT, Delta, Omicron BA.2), and 16 µg/ml (Omicron BA.1). Triplicate or duplicate preparations of each antibody dilution were incubated with the same volume of 100 $TCID_{50}$ of authentic SARS-CoV-2 WT (IME-BJ01 strain, GenBank No. MT291831), Beta (CSTR: 16698.06.NPRC2.062100001), Delta (CSTR.16698.06.NPRC6.CCPM-B-V-049-2105-6), Omicron BA.1 (SARS-CoV-2 strain Omicron CoV/human/CHN_CVRI-01/2022), and BA.2 (SARS-CoV-2 strain Omicron CoV/human/CHN_CVRI-04/2022) strains at 37 °C for 1 h. The mixtures were then transferred to 96-well plates seeded with Vero cells (ATCC, CCL-81, >80% density)[86]. After culturing at 37 °C for 4 days, CPEs caused by virus infection were scored for each well in a blinded fashion. The results were then converted into the percentage of neutralization at a given antibody concentration, and the averages ± s.d. (triplicates) or averages (duplicates) were plotted using nonlinear regression (log [inhibitor] vs. normalized response, variable slope) (GraphPad Prism v.8.0).

## SARS-CoV-2-specific memory B-cell analyses

PBMCs from vaccinated individuals were thawed and blocked with Human TruStain FcX Fc (Biolegend, 422302) for 10 min at 4 °C, followed by incubation in cell staining buffer (1× PBS, 2% FBS) with Biotinylated SARS-CoV-2 S protein (Acro Biosystems, SPN-C82E9), Biotinylated SARS-CoV-2 Spike Trimer (T19R, G142D, EF156-157del, R158G, L452R, T478K, D614G, P681R, D950N) (Acro Biosystems, SPN-C82Ec), or Biotinylated SARS-CoV-2 Spike Trimer (B.1.1.529/Omicron) (Acro Biosystems, SPN-C82Ee) for 60 min at 4 °C and then incubation for 30 min at 4 °C with the following anti-human antibodies (all at a 1:100 dilution): anti-CD19-FITC (Biolegend, 363008, Clone HIB19), anti-CD3-Pacific Blue (Biolegend, 300431, Clone UCHT1), anti-CD8-Pacific Blue (Biolegend, 301023, Clone RPA-T8), anti-CD14-Pacific Blue (Biolegend, 325616, Clone HCD14), anti-CD27-PerCP/Cyanine5.5 (Biolegend, 356408, Clone M-T271), streptavidin-APC (Biolegend, 405207) and streptavidin-PE (Biolegend, 405203). The $CD3^-CD8^-CD14^-CD19^+CD27^+$Spike-PE$^+$Spike-APC$^+$ B cells were quantified using a CytoFLEX LX (Beckman Coulter) and CytExpert (v.2.4) for analysis.

## Single-cell sequencing

The frequency distributions of the human V genes, CDR3 length and nucleotide SMH in individuals before immunization with the inactivated SARS-CoV-2 vaccine were characterized by single-cell sequencing. PBMCs from three participants in the cohort were selected to enrich B cells with a pan-B-cell isolation kit (Miltenyi Biotec, 130-101-638). After being blocked with Human TruStain FcX Fc (Biolegend, 422302) for 10 min at 4 °C, the enriched B cells from different

participants were stained with distinct TotalSeq C antibodies (all at 1:250 dilution), TotalSeq-C0251 Anti-Human Hashtag 1 Antibody (Biolegend, 394661), TotalSeq-C0252 Anti-Human Hashtag 2 Antibody (Biolegend, 394663), and TotalSeq-C0253 Anti-Human Hashtag 3 Antibody (Biolegend, 394665), and incubated in cell staining buffer (1× PBS, 2% FBS) for 30 min at 4 °C with the following anti-human antibodies (all at 1:100 dilution): anti-CD19-FITC (Biolegend, 363008, Clone HIB19), anti-CD3-Pacific Blue (Biolegend, 300431, Clone UCHT1), anti-CD8-Pacific Blue (Biolegend, 301023, Clone RPA-T8), anti-CD14-Pacific Blue (Biolegend, 325616, Clone HCD14), anti-CD27-PerCP/Cyanine5.5 (Biolegend, 356408, Clone M-T271). Single $CD3^-CD8^-CD14^-CD19^+CD27^+$ B cells were gated and sorted into Eppendorf tubes containing PBS with 10% FBS using a MA900 Cell Sorter (Sony).

Cells were counted and prepared for constructing 5′-mRNA, VDJ, and feature barcode libraries using the 10× Chromium System (10X Genomics) according to the manufacturer's instructions. The Chromium Next GEM Single Cell 5′ Kit v2 (10X Genomics, PN-1000266), Library Construction Kit (10X Genomics, PN-1000196), Chromium Next GEM Single Cell 5′ Gel Bead Kit v2 (10X Genomics, PN-1000267), 5′ Feature Barcode Kit (10X Genomics, PN-1000256), BCR Amplification Kit (10X Genomics, PN-1000253), Chromium Next GEM Chip K Single Cell Kit (10X Genomics, PN-1000287), Dual Index Kit TT Set A (10X Genomics, PN-1000215), and Dual Index Kit TN Set A (10X Genomics, PN-1000250) were used. All the libraries were quantified by using Fragment Analyzer (Agilent) and sequenced on a NovaSeq 6000 (Illumina) with 10 cycles for the i7 index and i5 index. The average sequencing depth aimed at the mRNA library was 20,000 read pairs per cell and 5000 read pairs per cell for the VDJ libraries and for feature barcode libraries.

### RBD-specific single B-cell sorting
B cells were enriched among PBMCs from vaccinated individuals using a pan-B-cell isolation kit (Miltenyi Biotec, 130-101-638) according to the manufacturer's instructions. The enriched B cells were blocked with Human TruStain FcX Fc (Biolegend, 422302) for 10 min at 4 °C, and incubated for 60 min at 4 °C in cell staining buffer (1× PBS, 2% FBS) with biotinylated SARS-CoV-2 spike RBD (Acro Biosystems, SPD-C82E9), and then incubated for 30 min at 4 °C with the following anti-human antibodies (all at a 1:100 dilution): anti-CD19-FITC (Biolegend, 363008, Clone HIB19), anti-CD3-Pacific Blue (Biolegend, 300431, Clone UCHT1), anti-CD8-Pacific Blue (Biolegend, 301023, Clone RPA-T8), anti-CD14-Pacific Blue (Biolegend, 325616, Clone HCD14), anti-CD27-PerCP/Cyanine5.5 (Biolegend, 356408, Clone M-T271), streptavidin-APC (Biolegend, 405207) and streptavidin-PE (Biolegend, 405203). Single $CD3^-CD8^-CD14^-CD19^+CD27^+RBD^+$ B cells were gated and sorted into 96-well PCR plates containing 4 μl of lysis buffer (0.5× PBS, 10 mM DTT, 10 units of RNase Inhibitor (New England Biolabs, M0314L)) per well using an MA900 Cell Sorter (Sony) for acquisition and Cell Sorter Software (v.3.1.1) for analysis. The plates were snap-frozen on dry ice and then immediately used for subsequent RNA reverse transcription or stored at −80 °C.

### Antibody amplification, cloning and expression
Human antibody heavy and light chain variable genes were generated as previously described[87]. RNA from single B cells was reverse transcribed using High-Capacity cDNA Reverse Transcription Kit (Thermo Fisher Scientific, 4368813), followed by nested PCR for amplification of the variable IGH, IGL and IGK genes. The products of the second round of PCR were purified and cloned into antibody expression vectors encoding the constant regions of human IgG1 by enzymatic assembly[88]. The IGBLAST program (https://www.ncbi.nlm.nih.gov/igblast/igblast.cgi) was used to analyze germline genes, germline divergence or the degree of somatic hypermutation (SHM), the framework region (FR) and the loop length of CDR3 for each antibody clone.

The paired heavy and light chain constructs were cotransfected into HEK293T cells grown in 12-well plates. The transfected culture supernatants were directly tested for binding and neutralization. Expression plasmids of mAbs showing neutralizing activity were transiently transfected into HEK293F cells (Thermo Fisher Scientific, R79007) with polyetherimide (PEI) (Polysciences, 24765) and the supernatant was purified using Protein G bead columns (Solarbio, R8300) according to the manufacturer's protocol. Variable gene sequences of anti-SARS-CoV-2 antibodies (CB6, C121, COV2-2130, COVA1-16, and ADG-2) were synthesized (Tsingke) and produced in a HEK293F cell system.

### Design, expression, and purification of bispecific antibodies
BsAbs were designed in the IgG-ScFv format. BI-2C5B and BI-5B2C were constructed from the sequences of the mAbs 6-2C and 10-5B. For BI-2C5B, the 10-5B ScFv was connected to the C-terminus of the 6-2C heavy chain with $(G4S)_5$ linkers to form the heavy chain, which was paired with the light chain of 6-2C. For BI-5B2C, the 6-2C ScFv was linked to the C-terminus of the 10-5B heavy chain with $(G4S)_5$ in the heavy chain, which was paired with the light chain of 10-5B. Similarly, BI-2C1C and BI-1C2C were constructed from the sequences of the mAbs 6-2C and 13-1C. The antibodies were produced by transient PEI transfection into HEK293F cells, and the supernatant was purified using Protein G bead columns according to the manufacturer's protocol.

### Biolayer interferometry (BLI)
Antibody affinity and the competitive binding of antibodies and hACE2 (or between two antibodies) were assessed using an Octet RED384 system (FortéBio). For apparent affinity ($K_D$ app) determination, 10 μg/ml recombinant biotinylated RBD of wild type (Acro Biosystems, SPD-C82E9), Omicron BA.1 (Acro Biosystems, SPN-C82E4), Omicron BA.2 (Acro Biosystems, SPN-C82Eq), and Omicron BA.4/BA.5 (Acro Biosystems, SPN-C82Ew) were loaded (1.0–1.2 nm) onto streptavidin biosensors (Molecular Devices, 18-5019), respectively. After a baseline step in PBS (Gibco, C10010500BT) for 60 s, the antigen-loaded biosensors were exposed to the mAbs for 200 s and then dipped (200 s) into PBS to measure any dissociation of antibodies from the biosensor surface. Data for which the binding responses were >0.1 nm were aligned, interstep corrected (to the association step) and fitted to a 1:1 binding model using FortéBio data analysis software, version 12.1.

For the hACE2 competition assay, 10 μg/ml recombinant biotinylated RBD (Acro Biosystems, SPD-C82E9) was immobilized (1.0–1.2 nm) onto streptavidin biosensors (Molecular Devices, 18-5019). After a baseline step in PBS buffer for 60 s, the mAbs (300 nM) were incubated with the RBD-coated biosensor for 300 s. After another baseline step in 1× PBS for 60 s, the biosensors were incubated with the hACE2 receptor (150 nM) (Sino Biological, 10108) for 300 s. The maximum binding of hACE2 was normalized to a PBS-only control. The percent binding of hACE2 in the presence of the antibody was compared to the maximum binding of hACE2. A reduction in the maximal signal to less than 20% was considered hACE2-blocking.

Epitope binning was performed with an in-tandem assay with streptavidin biosensors (Molecular Devices, 18-5019). The loaded biosensors were immersed in PBS for 60 s and then associated with the first antibody (Ab1, 300 nM) for 300 s. After a 60 s baseline step in 1× PBS, the sensors were associated with the second antibody (Ab2, 150 nM) for 300 s. Curve fitting was performed using FortéBio data analysis HT v12.1.

### In vivo efficacy in a humanized ACE2 mouse model
Studies with authentic SARS-CoV-2 Delta in a humanized ACE2 mouse model were performed in biosafety level 3 laboratories and approval was obtained from the Institutional Animal Care and Use Committees

of the Guangzhou Medical University. Female nine- to twelve-week-old humanized ACE2 mice were purchased from GemPharmatech (T037630). Animals were housed in a negative pressured isolator under 12 h light–dark cycles with a temperature at 22 °C and humidity set points of 50–60%. For the prophylactic experiment, the mice were intraperitoneally administered one dose of 20 mg/kg mAb 10-5B, 6-2C, or 10-5B + 6-2C (1:1 ratio) or PBS alone as a control. After 24 h, the mice were anesthetized and intranasally inoculated with $5 \times 10^4$ FFU of authentic SARS-CoV-2 Delta variant. Their body weight was measured daily, and the mice were euthanized to collect lung tissue at 3 days post infection (dpi).

The virus titers of the right lung homogenate were measured by FFU assay. Serially diluted homogenates were added to Vero E6 cells in 96-well plates and incubated for 1 h at 37 °C. Afterward, the inoculum was removed, and overlay medium (MEM containing 1.6% carboxymethylcellulose) was added. Twenty-four hours later, the cells were fixed with 4% paraformaldehyde and permeabilized with 0.2% Triton X-100. After incubation with a rabbit anti-SARS-CoV-2 nucleocapsid protein polyclonal antibody (Sino Biological, 40143-T62) at a 1:3000 dilution, the cells were labeled with an HRP-labeled goat anti-rabbit secondary antibody (Jackson ImmunoResearch Laboratories, 111-035-144) at a 1:10000 dilution. The foci were visualized with TrueBlue Peroxidase Substrate (KPL, 50-78-02) and counted with an ELISPOT reader (Cellular Technology).

### In vivo efficacy in a K18-hACE2 mouse model
Studies with authentic SARS-CoV-2 Omicron BA.2 (SARS-CoV-2 strain Omicron CoV/human/CHN_CVRI-04/2022) were performed in biosafety level 3 laboratories. Female seven to eight-week-old K18-hACE2 transgenic mice were purchased from GemPharmatech (T037657) and randomly allocated to groups. For the prophylactic experiment, the mice were intraperitoneally administered one dose of 10 mg/kg mAb BI-2C5B or PBS alone as a control. After 24 h, the mice were anesthetized and intranasally inoculated with $3.6 \times 10^3$ $TCID_{50}$ of authentic SARS-CoV-2 Omicron BA.2 variant. Their body weight was measured daily, and the mice were euthanized 3 days post infection (dpi) to harvest lung tissues for virological assessment or histological examination.

SARS-CoV-2 E gene sgRNA was quantified by quantitative reverse transcription-PCR (qRT–PCR) assays using primers and probes as previously described[89,90]. Briefly, RNA samples collected from challenged mice were reverse transcribed using HiScript III RT SuperMix (Vazyme, R323-01) followed by PCR using AceQ qPCR Probe Master Mix (Vazyme, Q112-02) with 400 nM concentrations of each of the primers, as well as 200 nM of probe (GenScript). Reactions were performed on a CFX96 Touch real-time PCR detection system (Bio-Rad). Standard curves were used to calculate the level of sgRNA in copies per gram.

### Histology
The lung tissues were fixed with neutral buffered formalin for 7 days before further processing. The tissues were embedded in paraffin, and sections were stained with hematoxylin and eosin[91]. Images were scanned using a Pannoramic MIDI (3D HISTECH). To detect the viral antigen, a rabbit anti-SARS-CoV/SARS-CoV-2 nucleocapsid antibody (Sino Biological, 40143-R004) at a 1:200 dilution was used, followed by incubation with an Alexa Fluor 488 goat anti-rabbit antibody (Wuhan Bioqiandu Technology Co., Ltd, B100805) at a dilution of 1:200. Images were acquired on a NIKON DS-U3 Imaging system and analyzed with Image-Pro Plus 6.0 (Media Cybernetics).

### Crystallization and data collection
The 6-2C Fab fragments were mixed with the SARS-CoV-2 spike RBD at a molar ratio of 1:1.2. Then, we purified the complex by gel-filtration chromatography. The purified complex was concentrated to 14 mg/ml

in HBS buffer (10 mM HEPES, pH 7.2, 150 mM NaCl) for crystallization. The crystallization temperature was set at 18 °C. The sitting drop vapor diffusion method was used by mixing 0.2 μl of protein with 0.2 μl of reservoir solution. Crystals of RBD-Fab complexes were successfully obtained in 0.1 M BIS-TRIS pH 6.5, 16% w/v polyethylene glycol 10,000. Diffraction data were collected at the BL02U1 beamline of the Shanghai Synchrotron Research Facility (SSRF) and processed with HKL2000 v721.4[92].

### Structural determination and refinement
Structures were determined by the molecular replacement method using PHASER in CCP4 Program Suite v7.0[93]. The search models were the SARS-CoV-2 RBD structure (PDB: 6M0J) and the heavy and light chain variable domain structures available in the PDB with the highest sequence identities. Subsequent model building and refinement were performed using COOT[94] and PHENIX[95], respectively. All structural figures were generated using PyMOL 2.0[96].

### Cryo-EM sample preparation and data collection
Aliquots of complexes (4 μl, in buffer containing 20 mM Tris, pH 8.0, and 150 mM NaCl) of the SARS-CoV-2 spike ectodomains (WT 1.65 mg/ml, BA.1 2 mg/ml, BA.4 1.3 mg/ml, respectively) and Fab (6-2C Fab, 10-5B Fab) were applied to glow-discharged holey carbon grids (Quantifoil grid, Cu 300 mesh, R1.2/1.3). The Fab fragments were mixed with the SARS-CoV-2 spike trimer at a molar ratio of 1.2:1. The grids were then blotted for 2 s and immediately plunged into liquid ethane using a Vitrobot Mark IV (Thermo Fisher Scientific). The cryo-EM data of the complexes were collected with an FEI Titan Krios microscope (Thermo Fisher Scientific) at 300 kV with a Gatan K3 Summit direct electron detector (Gatan Inc., Pleasanton, CA, USA) at Tsinghua University. A total of 5494 movies were collected with SerialEM v 4.0.4, with a magnification of 29000 and defocus range between −1.3 and −1.5 μm. Each movie had a total accumulated exposure of 50 e-/$Å^2$ fractionated in 32 frames of 2.13 s exposures. The stacks were binned twofold, resulting in a pixel size of 0.97 Å/pixel.

### Cryo-EM data processing
Motion correction (MotionCor2 v.1.2.6), CTF estimation (GCTF v.1.18), and nontemplated particle picking (Gautomatch v.0.56; http://www.mrc-lmb.cam.ac.uk/kzhang/) were automatically executed using the TsingTitan.py program[97,98]. Sequential data processing was performed in cryoSPARC v3.3.1[99,100]. Details of the data collection and processing are shown in Supplementary Figs. 13–16 and Supplementary Tables 6–7.

### Cryo-EM model building and refinement
The initial model of the SARS-CoV-2 spike ectodomains (PDB 7DWY) and 10-5B Fab was fit into the map using UCSF Chimera v.1.15[101]. Manual model rebuilding was performed with COOT v.0.9.2 and refined with PHENIX v.1.18.2 real-space refinement[94,95]. The quality of the final model was analyzed with PHENIX v.1.18.2[95]. The validation statistics of the structural models are summarized in Supplementary Table 7. All structural figures were generated using PyMOL 2.0 and Chimera v.1.15[96,101].

### Quantification and statistical analysis
No statistical methods were used to predetermine the sample size. The animals were randomly allocated into different groups. A description of the statistical analysis is provided in the figure legends. The analyses were performed using Prism 8.0 software (GraphPad, La Jolla, CA, USA). $p < 0.05$ was considered statistically significant.

### Reporting summary
Further information on research design is available in the Nature Portfolio Reporting Summary linked to this article.

## Data availability
The crystal structure presented in this work has been deposited in the Protein Data Bank (PDB) with accession code 7X2H for 6-2C/RBD. The cryo-EM structural data and models are available from the PDB under accession codes 7XD2 for 10-5B/WT S, 8H08 for 10-5B/6-2C/BA.1 S, and 8H07 for 10-5B/6-2C/BA.4 S. Sequences of the monoclonal antibodies characterized here have been deposited at Science Data Bank (https://doi.org/10.57760/sciencedb.07696). Source data are provided with this paper.

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

## Acknowledgements

We thank all the participants in our studies. We thank Prof. Linqi Zhang and Dr. Lingxuan Shi for sharing the plasmids and proteins. We acknowledge Mengsi Sun (the office of core facilities of Shenzhen Bay Laboratory) for his assistance with BLI assays, as well as Mengyuan Li and Jiying Hu for their help in flow cytometry. We are also grateful to Hao Guo and Songzhu Chen for their help with participants enrollment. We would also like to acknowledge Yibin Zhu for helpful suggestions. This work was supported by grants from the National Key R&D Program of China (2022YFC2303400 to Y.L.; 2021YFC2300200, 2020YFC1200104, 2018YFA0507202, 2021YFC2302405, 2022YFC2303200 to G.C.; 2021YFC2300104, 2022YFF1203103 to X.W.), the National Natural Science Foundation of China (32188101, 31825001, 81730063, and 81961160737 to G.C.; 32171202 to X.W.), the Yunnan Cheng gong expert workstation (202005AF150034 to G.C.), Innovation Team Project of Yunnan Science and Technology Department (202105AE160020 to G.C.), Tsinghua-Foshan Innovation Special Fund (2022THFS6124 to G.C.), Shenzhen San-Ming Project for prevention and research on vector-borne diseases (to G.C.), and Vanke Special Fund for Public Health and Health Discipline Development, Tsinghua University (20221080056 to X.W.).

## Author contributions

G.C. conceived and designed the study. G.C., X.W., Y.L., and Z.W. wrote and revised the manuscript. J.Z., M.T., W.S., and N.J. contributed experimental suggestions or helped with the work in BSL-3. Y.L., Z.W., X.Z., S.Z., Z.C., Y.Z., and J.S. performed most of the experiments. T.L., W.T., J.Y., Y.W., Z.Z., Y.C., L.T., X.Y., and L.W. helped with the experiments. Y.L. and Z.W. analyzed the data. D.C. and R.Z. helped to collect samples. All the authors reviewed, critiqued, and provided comments on the text.

## Competing interests

Patent applications have been filed that cover some of the antibodies presented here. G.C., Y.L., W.T., Y.Z., J.S., and Y.C. are inventors. The other authors declare no competing interests.

## Additional information

[1]Tsinghua-Peking Joint Center for Life Sciences, School of Medicine, Tsinghua University, Beijing 100084, China. [2]Institute of Infectious Diseases, Shenzhen Bay Laboratory, Shenzhen Guangdong 518132, China. [3]The Ministry of Education Key Laboratory of Protein Science, Beijing Advanced Innovation Center for Structural Biology, Beijing Frontier Research Center for Biological Structure, Collaborative Innovation Center for Biotherapy, School of Life Sciences, Tsinghua University, Beijing 100084, China. [4]Changchun Veterinary Research Institute, Chinese Academy of Agricultural Sciences, Changchun 130122, China. [5]State Key Laboratory of Respiratory Disease, Guangzhou Institute of Respiratory Disease, the First Affiliated Hospital of Guangzhou Medical University, Guangzhou 510182, China. [6]Center for Translational Research, Shenzhen Bay Laboratory, Shenzhen Guangdong 518132, China. [7]Wenzhou Central Hospital, Wenzhou 325000, China. [8]Institute of Pathogenic Organisms, Shenzhen Center for Disease Control and Prevention, Shenzhen Guangdong 518055, China. [9]These authors contributed equally: Yubin Liu, Ziyi Wang, Xinyu Zhuang, Shengnan Zhang, Zhicheng Chen. ✉e-mail: wshen@szbl.ac.cn; zhaojincun@gird.cn; klwklw@126.com; xinquanwang@mail.tsinghua.edu.cn; gongcheng@mail.tsinghua.edu.cn

