## [Peer Review File · Nature Communications]

Inactivated vaccine-elicited potent antibodies can broadly neutralize SARS-CoV-2 circulating variantsREVIEWER COMMENTS

Reviewer #1 (Remarks to the Author):

1. What are the noteworthy results?

This paper reports crucial findings on inactivated vaccine-induced broad-spectrum antibodies with broad neutralization activity against SARS-CoV-2 and the currently circulating variants. The authors identified antibodies, 6-2C and 10-5B, with a potency comparable to or greater than that of the approved antibodies, COV2-2130 and ADG-2, and together with their combinations, 6-2C and 10-5B antibodies, can effectively protect hACE2 transgenic mice SARS-CoV-2 Delta or Omicron BA.2 variants. Also, bsAb BI-2C5B antibodies designed based on 10-5B and 6-2C potently and broadly neutralized all the tested variants with better IC 50 values than its parental antibodies. The characterized antibodies have a prophylactic potential for the COVID-19 pandemic management.

2. Will the work be of significance to the field and related fields?

The characterized antibodies can guide future vaccine and therapeutic design and updates. The findings of this report should have been quite informative if immunologic shield after vaccination with the inactivated vaccine, that is, persistence and protective potential of antibodies and memory B cells, were evaluated and characterized beyond three (3) months.

3. How does it compare to the established literature?

The work in the manuscript is original. Previous studies were on SARS-CoV-2 inactivated vaccine enhanced broad neutralization against variants in individuals recovered from COVID-19 and heterologous immunization with inactivated vaccine followed by mRNA-booster elicits strong immunity against SARS-CoV-2 Omicron variant.

4. If the work is not original, please provide relevant references.

N/A

5. Does the work support the conclusions and claims, or is additional evidence needed?

Yes. The work supports the authors' conclusions and claims.

6. Are there any flaws in the data analysis, interpretation and conclusions?

The authors, however, should explain why antibodies against S, RBD, S2 and N viral components were present in pre-vaccinated study participants, considering that the study excluded those with a history of

COVID-19 diagnosis, see Fig 1b. Were the COVID-19 vaccinated participants excluded? This information is not clear from the methodology.

7. Do these prohibit publication or require revision?

No, they don't prohibit publication but require revision to address the minor issues raised.

8. Is the methodology sound? Does the work meet the expected standards in your field?

Yes, the methods are logically thought out and systematically applied, following the expected standards in molecular biology.

-correct referencing style (lines 736-7)

9. Is there enough detail provided in the methods for the work to be reproduced?

Yes, the methods are in detail.

10. Other suggestion

The title can be revised to: 'Inactivated vaccine-elicited potent antibodies can broadly neutralize SARS-CoV-2 circulating variants'.

Reviewer #2 (Remarks to the Author):

Liu et al., revealed that sera from vaccinees that received three doses of the BBIBP-CorV SARS-CoV-2 inactivated vaccine were tested for efficacy against VOC, VOI, and omicron variants. Subsequently, isolated antibodies were characterized in VDJ sequences and those capable of neutralizing a wide range of variants were found. Moreover, the two antibodies, 6-2C and 10-5B, derived from the serum after the second vaccination, were selected to prepare their bi-specific antibodies. Their structures determined by cryo EM or x-ray crystallography revealed both the antibodies recognized non-overlapped and distinct epitopes in ACE2 binding area, which is consistent with neutralizing activity of bi-specific antibodies toward a wide range of VOCs. In addition, their in vivo effect in a mouse model (Delta or Omicron BA.2 variant) showed a protective effect. The effectiveness of this inactivated vaccine was extensively tested on a range of variants, including Omicron. The study is extensive and interesting. I have some comments as follows.

1) Fig. 2a. Please mention how to prepare the samples of healthy donors and describe how efficient to find the RBD-specific B cells compared with vaccinees. Please discuss why VH/VL mutations are higher before vaccination than those after vaccination.

2) The authors selected 118 monoclonal antibodies (65 from dose 2 and 53 from dose 3). They further characterized 20 antibodies that harbored potent neutralizing activity (toward WT?). While the serum after the third vaccination showed more neutralizing activity than those after the second vaccination, the selected antibodies from dose 3 showed less breadth than those of dose 2 (Fig. 2d). Is there any possibility that the antibody selection might include any bias? If they use Omicron RBD for B cell selection, what do the authors expect? Please discuss. (The authors selected two antibodies, 6-2C and 10-5B, derived from the serum after the second vaccination, to be analyzed further afterward based on the viral breadth. Especially, 10-5B exhibited potent neutralizing activity toward Omicron variants (BA.1, BA.2, BA.3).)

3) The refinement parameters of the crystal structure of 6-2C complexed with RBD seems fine, while there are some outliers in the PDB validation report. On the other hand, the percent of the structural parameter outliers of the EM structures seems high probably due to low-resolution data. Therefore, the authors should carefully evaluate the structures and show electron density maps for the RBD-Fab binding interfaces. They might better not discuss the detailed interactions.

4) Discussion: Please discuss the relationship between the preference of VH/VL gene preference (lines 387-391) and structural features (lines 402-406).

5) There exist two epitope classifications, RBD1-7 and Class 1-4. Please describe both or use one.

6) Fig. 3b: Please use "apparent" KD in Fig. 3b, because the authors use IgG (two binding sites, 1:2 binding, avidity effect) not Fab.

7) Fig. 4b: The body weight changes (Delta) in control were not sufficiently large. Are there any reasons?

8) Please describe in detail each panel of Supplementary Fig. 3.

Reviewer #3 (Remarks to the Author):

Liu et al., characterize the human antibody response to inactivated, whole virus, SARS-CoV-2 vaccines. From 118 monoclonal antibodies recovered from 28 human donors, 20 showed impressive breadth of binding. Two antibodies, directed to distinct epitopes, were selected to produce bi-specific antibodies. These engineered antibodies have demonstrably improved avidity for SARS-CoV-2 Spike and greater resistance to antigenic variation present in Omicron sub-lineages. These may provide therapeutic advantages as Omicron sub-lineages have evaded all current therapeutic and prophylactic antibody therapies. The work spans antibody repertoire analysis, viral neutralization assays, the generation of bi-specific antibodies, structure determination, and in vivo protection data in rigorous animal models. The work is expertly performed and highly relevant to the current dearth of monoclonal antibody products.

The reported structures, given their resolution, have very high MolProbity scores and an unusual percentage of unfavored rotamers. These structures need to be improved before publication.

The authors should specify in the text of the manuscript what the comparison to “control” antibodies represents (e.g. Figure 2). Specifically, how these antibodies were isolated, why these represent controls and to clarify what control population of B cells represents in Figure 2 would be helpful.

The criteria to select antibody candidates for bi-specific antibodies is not clearly defined. Antibody 13-1C has remarkable breadth of binding but was not chosen for bi-specific antibody optimization.

Contextualization of the findings of this manuscript with those of the Bloom group and others should be discussed more rigorously.

Minor:

The use of the same color scheme to denote vaccine doses (e.g. does 1 and dose 2), spike binding affinity (Figure 2) and beyond (e.g. Figure 2e) are confusing.

Lines 156-161 could be stated more clearly.

Discussion of reported antibody germline associations with specific epitopes does not necessarily predict antibody epitopes. Without competition data, discussion of germline associations should be curtailed.

Reviewer #4 (Remarks to the Author):

In this paper titled “Potent neutralizing antibodies elicited by an inactivated vaccine can broadly resist SARS-CoV-2 circulating variants” Liu, Cheng and colleagues report on the polyclonal antibody responses and identification of broadly neutralizing and protective monoclonal antibody elicited by BBIBP-CorV vaccination. The objective of the study was to determine cross-neutralization and protection of SARS-CoV-2 variants after one, two or (months later) after three boosts with the inactivated BBIBP-CorV vaccine. At the serological level, the authors found that plasma neutralizing titers against SARS-CoV-2 subvariants, including Omicron variants, were enhanced after the third dose compared to after the second dose of inactivated vaccine. Serum antibody levels to the variants roughly correlated to frequencies of memory B cells assayed by flow cytometry and using fluorescent spike from variants as probes, though only a subset of subjects were assayed and this number diminished for later variants. From the memory B cells they found distinctions in Ig repertoire such as IgV genes used, CDR3 lengths and accumulating mutations. 118 mAbs were generated from memory B cell Ig genes after the second or third boosts. 20/118 mAbs showed neutralizing activity and 3 had broad variant cross-neutralizing activity and showed in vivo protection. Structural analysis revealed the target epitope of these broadly neutralizing antibodies were on RBD communities’ sites (RBD2 and RBD5-7). By analyzing monoclonal antibodies established from vaccinated individuals, antibodies isolated after second dose and third dose displayed distinct neutralization profiles and epitope specificity. Furthermore, bispecific antibodies were generated using neutralizing antibodies with non-overlapping epitopes, and their neutralizing activity was shown to be stronger and broader.

1. Information critical to the interpretation of this study is to provide either timing of sample collection prior to the variants arising or other evidence that none of the subjects were infected to the variants possibly asymptotically, accounting for cross-variant activities.
2. For accuracy, please add a description of the control specimen to the main text. Are the controls pre-vaccinated samples of BBIBP-CorV vaccinated individuals?
3. It should be clearly and explicitly stated that only a subset of subjects were studied for memory B cells, Ig repertoire, and mAbs. Further, why was the subset of subjects from which memory B cells and mAbs produced chosen? Are these subjects representative or were they high responders? This is an important distinction.
4. Figures 1d-1f. In the sample one month after the third vaccination (Dose 3 + 1 Mo), only four individuals show high percentages of Spike-specific memory B cell. Can the authors explain the reason from their individual profiles?
5. Figure 2a. Are there differences in variable gene usage or SHM levels when comparing antibodies established from 1 month, 3 months, or 6 months after the second vaccination?
6. Are distinctions in the IgV gene repertoire (V gene usage, CDR3 length, etc) due to expansions of particular, related B cell clones? Do these same differences persist if each clone is considered as a single data point?
7. What is the significance of the changes in CDR3 length?

8. Figure 2d: The authors identify 3 broadly neutralizing mAbs against all tested variants, 2 of which were from the 3rd dose. However, only antibodies (10-5B and 6-2C) that derived from dose 2 were selected to further characterization (CPE, generation of bi-Abs, protective efficacy and structural analysis) whereas the 10-5B mAb was not able to neutralize BA.4/5. What is the rationale of selection 10-5B and 6-2C instead of 13-1C/ 13-1F paired with 6-2C since the footprint of 13-1C/13-1F also non-overlapping with 6-2C and displayed broadly neutralizing activity than 10-5B?
9. There should be improved clarity regarding from which boost and subjects the neutralizing mAbs arose. Were these distributed across the cohort or from only a few subjects? Were these mostly from the 3rd boost? This should be clear as mAbs lose activity to variants finally resulting in only 3 with broad-variant activities.
10. Please add information such as CDR3 sequence, SHM level, and variable gene for the 20 neutralizing antibodies to the table in Figure 2 for reference.
11. Figure 2e. Please indicate inside the figure which color means competition.
12. Line 196. The authors mentioned that antibodies isolated from samples collected after Dose 2 and Dose 3 had differential competition profiles and exhibited bias in epitope specificity. To further clarify this, the competitive ELISA with class I, II, III and IV antibodies should be performed on sera after the first, second, and third vaccine doses.

Minor:

1. As not all people are familiar with the various vaccines available, please provide a brief description of BBIBP-CorV (company, method of inactivation, geographical usage, etc.).
2. Line 100, grammar error, "were" not needed.

Responses to Reviewer #1:

#1A. *The authors, however, should explain why antibodies against S, RBD, S2 and N viral components were present in pre-vaccinated study participants, considering that the study excluded those with a history of COVID-19 diagnosis, see Fig 1b. Were the COVID-19 vaccinated participants excluded? This information is not clear from the methodology.*

#1A Answer: We recognize the reviewer's concern. Our exclusion criteria included COVID-19 vaccinated participants, and this information has been added in the Methods section and highlighted (**lines 524-525, page 20**). As for the antibodies against S, RBD, S2 and N protein in prevaccinated study participants, multiple independent assays demonstrated the presence of preexisting antibodies recognizing SARS-CoV-2 in uninfected individuals (**Ng, et al. *Science*. 2020,370:1339-1343; Cho, et al. *Sci Transl Med*. 2021,13:eabj5413.**). These antibodies were probably induced by other coronaviruses, particularly seasonally spreading human coronaviruses (HCoV), HCoV-HKU1 and HCoV-OC43, and vHCoV-NL63 and HCoV-229E, and cross-reacted with SARS-CoV-2 spike and nucleocapsid proteins (**Anderson, et al. *Cell*. 2021,184:1858-1864; Shrwani, et al. *J Infect Dis*. 2021, 224:1305-1315.**). We have added this explanation in the revised manuscript (**lines 95-99, page 5**).

#1B. *Correct referencing style (lines 736-7)*

#1B Answer: Thank you for the reminder. The reference style was corrected and highlighted in the revised manuscript (**line 803, page 30**).

#1C. *The title can be revised to: 'Inactivated vaccine-elicited potent antibodies can broadly neutralize SARS-CoV-2 circulating variants'.*

#1C Answer: Thank you for the suggestion. The title has been revised to "Inactivated vaccine-elicited potent antibodies can broadly neutralize SARS-CoV-2 circulating variants".

Responses to Reviewer #2:

#2A. *Fig. 2a. Please mention how to prepare the samples of healthy donors and describe how efficient to find the RBD-specific B cells compared with vaccinees. Please discuss why VH/VL mutations are higher before vaccination than those after vaccination.*

#2A Answer: We recognize the reviewer's concern. The preparation of samples of healthy donors has been added to the Methods section (**lines 602-630, pages 23-24**). Briefly, memory B cells of prevaccination samples of three participants in our cohort were sorted, and V genes, CDR3 length and SMH were characterized by single-cell sequencing. Paired heavy and light chains of IgG B cell receptors (BCRs) (n=403) from healthy donors represent baseline circulating IgG memory B cell repertoire (**Liu, et al. *Nature*. 2020,584:450-456; Seow, et al. *Cell Rep*. 2022,39:110757.**). This information has been described in the main text (**lines 151-**

156, page 7). Additionally, in the revised manuscript, the isotype of the controls was considered, and the data in **Fig. 2a-c** and **Supplementary Fig. 4** were revised.

We found that RBD-binding memory B cells had low SHM levels in VH and VL genes and were less mutated than those from healthy IgG⁺ repertoires, in line with the near-germline IgG antibodies against SARS-CoV-2 in natural infection, suggesting that these cells were primed by the ongoing vaccination and differentiated from B cells without extensive germinal center experience (*Liu, et al. Nature. 2020, 584:450-456; Kreer, et al. Cell. 2020,182:843-854; Cho, et al. Sci Transl Med. 2021,13:eabj5413; Rogers, et al. Science. 2020, 369:956-963.*). Notably, the timing of mAbs isolation following vaccination should be considered, as the SHM level increased over time both in our work and previous reports (*Gaebler, et al. Nature. 2021,591: 639-644; Wang, et al. Nature. 2021,595:426-431.*), indicating continued development of the humoral response. We have added this discussion in the revised manuscript (**lines 422-429, pages 16-17**).

#2B. *The authors selected 118 monoclonal antibodies (65 from dose 2 and 53 from dose 3). They further characterized 20 antibodies that harbored potent neutralizing activity (toward WT?). While the serum after the third vaccination showed more neutralizing activity than those after the second vaccination, the selected antibodies from dose 3 showed less breadth than those of dose 2 (Fig. 2d). Is there any possibility that the antibody selection might include any bias? If they use Omicron RBD for B cell selection, what do the authors expect? Please discuss. (The authors selected two antibodies, 6-2C and 10-5B, derived from the serum after the second vaccination, to be analyzed further afterward based on the viral breadth. Especially, 10-5B exhibited potent neutralizing activity toward Omicron variants (BA.1, BA.2, BA.3).)*

#2B Answer: We recognize the reviewer's concern. Twenty antibodies were selected with potent neutralizing activity toward WT. We compared the neutralizing mAbs obtained after the second dose and the booster dose. The average breadth of mAbs from dose 3 (54.4%) was slightly higher than that of those from dose 2 (49.8%) when we calculated the percentage of pseudoviruses (n=29) to which the mAbs had an IC₅₀ value below 10,000 ng/ml. However, the number of neutralizing mAbs was relatively small and might not be sufficient to analyze the difference. Another possible reason is that the mAbs were isolated from a subset (n=6) of subjects. Therefore, bias may exist because not all samples were equally evaluated. If Omicron RBD is used for B cell isolation, we expect the mAbs to exhibit enhanced breadth. WT vaccines-elicited antibodies by that bind and neutralize Omicron probably target conserved sites shared with the WT strain (*Goel, et al. Cell. 2022,185:1875-1887; Cameroni, et al. Nature. 2022, 602: 664-670.*). Additionally, screening for binding activity with the transfection supernatants, we found that Omicron RBD-binding mAbs simultaneously bound the WT, Alpha, Beta, and Delta strains. Nevertheless, studies are warranted to further investigate this possibility. We have added this discussion in the revised manuscript (**lines 430-443, page 17**).

#2C. *The refinement parameters of the crystal structure of 6-2C complexed with RBD seems fine, while there are some outliers in the PDB validation report. On the other hand, the percent of the structural parameter outliers of the EM structures seems high probably due to low-resolution data. Therefore, the authors should carefully evaluate the structures and show electron density maps for the RBD-Fab binding interfaces. They might better not discuss the detailed interactions.*

#2C Answer: Thank you for the suggestion. We improved the cryo-EM structures of the spike trimer bound with antibodies by optimizing the refinement, and the density maps of the RBD-Fab region were further improved by local refinement. The density maps of the 6-2C binding interfaces allow for model building of the side chains of most interacting residues. However, the density maps of the 10-5B binding interfaces were still less favored for model building of the side chains of interacting residues. Therefore, we stated that the numbers calculated by PISA at the 10-5B/RBDs were approximations, but we believed that they still provided rational information for investigating structural basis of binding and neutralization of 10-5B against Omicron subvariants. All these modifications were included in the revised manuscript (**lines 304-366, pages 12-14; Fig. 5d**) and the density maps of the RBD-Fab binding interfaces are shown in the **Supplementary Fig. 16**.

#2D. *Discussion: Please discuss the relationship between the preference of VH/VL gene preference (lines 387-391) and structural features (lines 402-406).*

#2D Answer: Thank you for the suggestion. We found that all nine IGHV3-53/IGHV3-66-derived antibodies of 20 neutralizing mAbs competed with class 1 and class 2 of RBD-targeting antibodies, which was consistent with the shared footprint on RBD and the similar angle approaching the RBD compared to that of ACE2 (**Barnes, et al. Nature. 2020,588:682-687; Barnes CO, et al. Cell. 2020,182:828-842; Zhang, et al. Nat Commun. 2021,12:4210.**). In addition, IGHV3-30 and IGHV1-69 gene-encoded neutralizing mAbs were classified into Class 2 and Class 3. Similarly, IGHV1-58 was enriched in mAbs isolated from Dose 3, and IGHV1-58 gene-encoded neutralizing tended to bind the left shoulder of RBD, often focusing on the far tip (**Cao, et al. Nature. 2022,602:657-663.**). Nevertheless, no obvious relationship between VL gene preference and structural features was observed in our study, possibly due to the relatively low contribution of the light chain to the epitope binding. We have added this discussion in the revised manuscript (**lines 465-475, page 18**).

#2E. *There exist two epitope classifications, RBD1-7 and Class 1-4. Please describe both or use one.*

#2E Answer: Thank you for the suggestion. In the competitive experiment, we used Class 1-4 to describe the epitope specificities of the neutralizing mAbs. As for the structural analysis, we have added the epitope classifications of Class 1-4 for the antibodies 6-2C and 10-5B (**lines 313-315, 321, page 13**) and mentioned the binding regions of antibodies based on the RBD1-7 classification.

#2F. *Fig. 3b: Please use “apparent” KD in Fig. 3b, because the authors use IgG (two binding sites, 1:2 binding, avidity effect) not Fab.*

#2F Answer: Thank you for the suggestion. Apparent dissociation constants ($K_{D, app}$) have been used in **Fig. 3b** and **Supplementary Fig. 11**.

#2G. *Fig. 4b: The body weight changes (Delta) in control were not sufficiently large. Are there any reasons?*

#2G Answer: We recognize the reviewer’s concern. One possible reason is that the humanized ACE2 mouse model we used (GemPharmatech, T037630) in Delta challenge is not sensitive to body weight loss, as the same Delta strain caused sufficient weight loss in a hACE2 transduced mouse model. We consulted more literature references, but weight change data was not available in other studies using the same mouse model (*Ma, et al. Immunity. 2020, 53: 1315-1330; Pan, et al. BioRxiv. 2020.10.14.335893*)

#2H. *Please describe in detail each panel of Supplementary Fig. 3.*

#2H Answer: Thank you for the suggestion. Each panel has been described in detail in the legend and highlighted.

Responses to Reviewer #3

#3A. *The reported structures, given their resolution, have very high MolProbity scores and an unusual percentage of unfavored rotamers. These structures need to be improved before publication.*

#3A Answer: Thank you for the suggestion. We have further improved the structures of these three complexes (WT Spike&10-5B, BA.1 Spike&105B&6-2C and BA.4 Spike&10-5B&6-2C) and updated the **Supplementary Table 7** in the revised manuscript. The unfavored rotamers percentages of three structures have been reduced to less than 1.2.

#3B. *The authors should specify in the text of the manuscript what the comparison to “control” antibodies represents (e.g. Figure 2). Specifically, how these antibodies were isolated, why these represent controls and to clarify what control population of B cells represents in Figure 2 would be helpful.*

#3B Answer: Thank you for the suggestion. The “control” antibodies represent the baseline circulating IgG-expressing memory B cell repertoire, which was isolated from prevaccination samples of three participants in our cohort, and the V genes, CDR3 length and SMH were characterized by single-cell sequencing. This information has been specified in the text (**lines 151-156, page 7**). The preparation of samples of healthy donors were mentioned in the Methods section (**lines 602-630, pages 23-24**). Additionally, in the new manuscript, the isotype of the controls was considered, and the data in **Fig.2a-c** and **Supplementary Fig. 4** have been revised.

#3C. *The criteria to select antibody candidates for bi-specific antibodies is not clearly defined. Antibody 13-1C has remarkable breadth of binding but was not chosen for bi-specific antibody optimization.*

#3C Answer: Thank you for the question. The characterization of 13-1C (neutralizing activity against live virus, generation of bsAbs) has been added in the revised manuscript since the COVID-19 pandemic prevented us from undertaking further work on the mAbs from dose 3 (**Fig. 2f, Fig. 3, and Supplementary Figs. 9-12**). The results revealed that 13-1C derived bsAbs, BI-2C1C and BI-1C2C, also exhibited more potent and broadly neutralizing activity against pseudoviruses than the parental mAbs and showed comparable neutralizing capacity to that of BI-2C5C and BI-5B2C (**Fig. 3e**). However, no matter 13-1C or BI-2C1C and BI-1C2C displayed much lower inhibitory activity against the authentic viruses (WT, Beta, Delta, BA.1, and BA.2) than that of 10-5B and its derived bsAbs in vitro (**Fig. 2f, Fig. 3f, and Supplementary Fig. 9**). Therefore, the protective efficacy and structural analysis of 13-1C and the bsAbs were not further examined in this work.

#3D. *Contextualization of the findings of this manuscript with those of the Bloom group and others should be discussed more rigorously.*

#3D Answer: Thank you for the suggestion. 6-2C broadly neutralized all the tested variants, including the most antibody-evasive variants XBB.1.5, XBB, BQ.1.1, and BA.2.75.2. In general, core-RBD epitopes tend to be mutationally constrained with respect to folding and expression and are mostly conserved across sarbecoviruses, explaining the possible neutralization breadth of antibody 6-2C (***Starr, et al. Cell. 2020,182:1295-1310.***). This discussion has been added in the revised manuscript (**lines 323-327, page 13**).

Furthermore, the distinct neutralization profiles between mAbs isolated after the second vaccine dose and the booster dose broadly corresponded to the difference in the competitive pattern of these mAbs. Four of seven mAbs (1-1D, 1-2D, 2-2E, 3-7D) obtained from Dose 2 presented impaired neutralizing activity against variants carrying the N501Y and/or E484K/A mutation, which could be explained by their competition binding with Class 2 antibody (C121). This finding was consistent with prior reports showing that the E484K mutation was associated with resistance to class 2 mAbs (***Greaney, et al. Nat Commun. 2021,12:4196; Wang, et al. Immunity. 2021,54:1611-1621.***). Although 1-1D and 1-2D displayed better tolerance to single E484K/A mutation, it rendered them inactive when combined with N501Y. Unlike the neutralizing mAbs isolated from Dose 2, nearly half of the neutralizing mAbs isolated after Dose 3 were more susceptible to variants with the L452R/Q mutation (Delta, Lambda, Kappa, Epsilon, et al), and epitope analysis revealed that all these antibodies competed with the Class 3 antibody (COV2-2130), supporting previous findings showing that L452 substitutions escaped Class 3 antibodies (***Cao, et al. Nature. 2022,608:593-602.***). Additionally, as revealed by deep mutational scanning, mutations at F486 also had substantial antigenic effects on antibodies, which rendered the candidate antibody 10-5B inactive due to breaking the hydrogen bond network around the E484-Y489 region. (***Greaney, et al. Cell Host Microbe.***

2021,29:463-476.). Furthermore, another promising antibody, 13-1C, was extensively affected by the new Omicron subvariants with the convergent mutation N460K in the RBD (**Cao, et al. Nature. 2022,10.1038/s41586-022-05644-7.**). Finally, the potency of the broadly neutralizing mAb 6-2C was impaired by the additional mutations R346T and K444T in RBD, similar to the neutralization features of class 3 mAbs (**Wang, et al. Cell. 2022,S0092-8674(22)01531-8**). We have added this discussion in the revised manuscript (**lines 444-465, pages 17-18**).

#3E. *The use of the same color scheme to denote vaccine doses (e.g. does 1 and dose 2), spike binding affinity (Figure 2) and beyond (e.g. Figure 2e) are confusing.*

#3E Answer: We recognize the reviewer's concern. The color scheme has been varied in **Figure 2**.

#3F. *Lines 156-161 could be stated more clearly.*

#3F Answer: Thank you for the suggestion. Lines 156-161 have been revised to read "The levels of SHM in the VH and VL genes of vaccinated individuals were significantly lower than those in healthy donors, and the mutation percentages increased after Dose 3 compared to those after Dose 2 (Fig. 2c). Furthermore, when mAbs from 1 month, 3 months, or 6 months after the second vaccination were stratified, an increase in SHM levels was observed over time for both the VH and VL genes (Supplementary Fig. 4d). Taken together, there was biased usage of IGHV, IGKV and IGLV genes in the vaccinated individuals, and a booster dose correlated with evolution of the memory B-cell compartment." (**Lines 168-176, page 8**).

#3G. *Discussion of reported antibody germline associations with specific epitopes does not necessarily predict antibody epitopes. Without competition data, discussion of germline associations should be curtailed.*

#3G Answer: Thank you for the suggestion. We have revised the discussion and removed the sentence "The features of these germline genes may provide insight for rational vaccine design involving the development of immunogens that can induce protective antibody responses specific to this site." (**Lines 414-429, pages 16-17**).

Responses to Reviewer #4

#4A. *Information critical to the interpretation of this study is to provide either timing of sample collection prior to the variants arising or other evidence that none of the subjects were infected to the variants possibly asymptotically, accounting for cross-variant activities.*

#4A Answer: We recognize the reviewer's concern. All samples were collected between May 2021 and May 2022. In China, the government adopted a series of strict prevention and control measures, including national SARS-CoV-2 nucleic acid testing, isolation and medical observation for close contacts, and early reporting of asymptomatic infections, before December 2022 (**Han, et al. Lancet. 2022,399:1690-1691; Li, et al. Nat Med. 2021, 27:740-742; Chen, et al. Int J Biol**

Sci. 2021,17:1119-1124.) Therefore, none of the subjects were infected during the period of our visit.

#4B. *For accuracy, please add a description of the control specimen to the main text. Are the controls pre-vaccinated samples of BBIBP-CorV vaccinated individuals?*

#4B Answer: We appreciate the suggestion. The control specimen was isolated from prevaccination samples of three participants in our cohort, which has been described in the main text (**lines 151-156, page 7**). The preparation of samples from healthy donors was described in the Methods section (**lines 602-630, pages 23-24**).

#4C. *It should be clearly and explicitly stated that only a subset of subjects were studied for memory B cells, Ig repertoire, and mAbs. Further, why was the subset of subjects from which memory B cells and mAbs produced chosen? Are these subjects representative or were they high responders? This is an important distinction.*

#4C Answer: We recognize the reviewer's concern. RBD-binding memory B cells were isolated from the blood of six individuals with top neutralizing activity after the second dose. To determine whether there were changes in the antibodies produced by memory B cells after the third dose, we obtained antibodies after Dose 3 from the same 6 individuals. The detailed information has been clearly and explicitly stated in the revised manuscript (**lines 146-160, page 7**) and **Supplementary Table 3**.

#4D. *Figures 1d-1f. In the sample one month after the third vaccination (Dose 3 + 1 Mo), only four individuals show high percentages of Spike-specific memory B cell. Can the authors explain the reason from their individual profiles?*

#4D Answer: Thank you for the suggestion. The four individuals included IDs 8, 11, 16, and 27, and their plasma samples also exhibited high anti-spike/RBD IgG titers and high neutralization titers after the third vaccination (**Table R1**). First, there was no significant difference in the factors (age, sex, BMI, and time since vaccination) associated with high antibody levels between the four individuals and other participants (*Wang, et al. Front Immunol. 2022,13:96705; Zhai, et al. J Immunol.2022,208:1711-1718.*). Additionally, their plasma activity and percentages of Spike-specific memory B cell before the third vaccination showed distinct levels. Finally, it was difficult to analyze the samples at the molecular level since only ID 11 was among the six individuals for mAbs isolation. Thus, the higher percentages of Spike-specific memory B cell in these individuals may be attributed to the inter- and intra-person heterogeneity as the circulating B cell populations of individuals are highly individualized and extremely diverse (*Briney, et al. Nature. 2019,566:393-397.*). Additionally, antibody responses varied substantially in convalescent and vaccinated individuals (*Wu F, et al. JAMA Intern Med. 2020,180:1356-1362; Ni, et al. Immunity. 2020,52:971-977; Sahin, et al. Nature. 2021,595:572-577.*) Similarly, there was extensive person-to-person variation in how mutations affect plasma antibody binding and neutralization (*Greaney, et al. Cell Host Microbe. 2021,29:463-476.*).

Table R1 Characteristics of subjects with high percentages of Spike-specific memory B cells after the booster dose

Subject ID	Age (years)	Sex	Samples obtained at	ELISA Binding (titer)		Pseudovirus neutralization (ID ₅₀)			
				Anti-Spike IgG	Anti-RBD IgG	WT	Previous VOCs (GMT)	Previous VOIs (GMT)	Omicron subvariants (GMT)
8	24	M	Dose 2 + 1M	8100	2700	21	14	8	6
			Dose 2 + 3M	900	300	5	n.d.	n.d.	n.d.
			Dose 2 + 6M	900	300	21	n.d.	n.d.	n.d.
			Dose 3 + 1M	24300	24300	766	192	330	27
11	33	M	Dose 2 + 1M	2700	2700	22	67	45	8
			Dose 2 + 3M	900	300	5	n.d.	n.d.	n.d.
			Dose 2 + 6M	300	300	5	n.d.	n.d.	n.d.
			Dose 3 + 1M	24300	8100	255	108	175	40
16	33	F	Dose 2 + 1M	8100	8100	61	29	25	9
			Dose 2 + 3M	2700	2700	19	n.d.	n.d.	n.d.
			Dose 2 + 6M	2700	900	10	n.d.	n.d.	n.d.
			Dose 3 + 1M	72900	72900	3120	643	547	24
27	23	M	Dose 2 + 1M	8100	2700	18	22	9	6
			Dose 2 + 3M	2700	900	10	n.d.	n.d.	n.d.
			Dose 2 + 6M	2700	900	20	n.d.	n.d.	n.d.
			Dose 3 + 1M	72900	72900	1117	405	255	18

GMT, geometric mean titer; M, male; F, female.

n.d., not determined.

#4E. *Figure 2a. Are there differences in variable gene usage or SHM levels when comparing antibodies established from 1 month, 3 months, or 6 months after the second vaccination?*

#4E Answer: Thank you for the question. We stratified antibodies established from 1 month (n=12), 3 months (n=38), or 6 months (n=15) after the second vaccination and compared their repertoire features. No significant differences were observed in V gene usage among them except that IGHV1-46 frequency was relatively higher in Dose 2 + 1 M than that in Dose 2 + 3 M. Notably, the number of IgK (n=5, n=26, n=14, respectively) and Igλ (n=7, n=12, n=1, respectively) sequences was small after being stratified, so the differences were not statistically significant although the frequencies were relatively high. However, the level of somatic hypermutation for both IGH and IGL increased significantly over time, indicating continued development of the humoral response. These findings were described in **lines 171-173** and figures were added in **Supplementary Fig. 4**.

#4F. *Are distinctions in the IgV gene repertoire (V gene usage, CDR3 length, etc) due to expansions of particular, related B cell clones? Do these same differences persist if each clone is considered as a single data point?*

#4F Answer: Thank you for the question. In this work, 33 clonally related sequences (same V and J genes at Heavy and Light Chains and similar CDR3) represents 16 expanded clones and comprised 28% of the RBD-specific antibodies. When each clone was considered as a single data point, V genes, such as IGHV3-30, IGHV3-53, IGHV3-13, as well as IGKV1-39, IGLV3-21 and IGLV6-57, were still significantly overrepresented in the RBD-binding memory B-cell compartment of vaccinated

individuals. Similarly, the difference in CDR3 length also persisted (**Fig. R1**). These findings were consistent with the results of a previous report in convalescent individuals, in which individual clones were covered and analyzed. They found “significant over-representation of IGHV3-30, IGKV3-20, and IGHJ6 genes for this collection of SARS-CoV-2 mAbs. In addition, the average CDRH3 length was also longer. Notably, the average percentages of somatic hypermutation in VH and VL were 2.1 and 2.5, respectively, which were significantly lower than those found in healthy individuals.” (*Liu, et al. Nature. 2020,584:450-456.*)

Fig. R1 Characterization of V genes from individual clones in anti-SARS-CoV-2 RBD monoclonal antibodies.

(a-c) The graph shows the relative abundance of human IGHV (a), IGVK (b) and IGLV (c) genes of anti-SARS-CoV-2 RBD monoclonal antibodies (mAbs) from vaccinees, compared to those in IgG-expressing memory B repertoires of healthy human donors. Statistical significance was determined by Chi-square test with 1 degree of freedom (* $p < 0.05$, ** $p < 0.01$, *** $p < 0.001$, **** $p < 0.0001$).

(d) The amino acid (aa) length of the CDR3 at IGHV and IGLV in mAbs as in a-c. The horizontal bars indicate the mean values. Statistical significance was determined by two-sided Kruskal Wallis test with subsequent Dunn's multiple comparisons.

#4G. *What is the significance of the changes in CDR3 length?*

#4G Answer: Thank you for the question. Long CDR3 length, especially long heavy chain CDR3 (HCDR3) length, is crucial for high-affinity antigen–antibody interactions (*Wu, et al. Nat Commun. 2017,8:15371.*). Long CDR3 length facilitates interaction with epitopes that are otherwise occult because of extensive glycosylation or location in recessed structures on the pathogen surface. Antibodies containing long HCDR3 loops have been shown to efficiently neutralize a wide variety of pathogens, including HIV, malaria, and African trypanosomes. (*Sok, et al. Nat Immunol. 2018,19:1179-1188; Sok, et al. Immunity. 2016,45:31-45; Briney, et al. PLoS One. 2012,7:e36750; Yu, et al. Front Immunol. 2014,5:250.*). Prior studies reported that the length of the HCDR3 was similar in COVID-19-convalescent individuals at 1.3, 6.2 and 12 months after infection (*Wang, et al. Nature. 2021,595:426-431.*). Similarly, we attempted to elucidate the features of CDR3 length after vaccination over time.

#4H. *Figure 2d: The authors identify 3 broadly neutralizing mAbs against all tested variants, 2 of which were from the 3rd dose. However, only antibodies (10-5B and 6-2C) that derived from dose 2 were selected to further characterization (CPE, generation of bi-Abs, protective efficacy and structural analysis) whereas the 10-5B mAb was not able to neutralize BA.4/5. What is the rationale of selection 10-5B and 6-2C instead of 13-1C/ 13-1F paired with 6-2C since the footprint of 13-1C/13-1F also non-overlapping with 6-2C and displayed broadly neutralizing activity than 10-5B?*

#4H Answer: We recognize the reviewer's concern. The characterization of 13-1C (neutralizing activity against live virus, generation of bsAbs) has been added in the revised manuscript, as the COVID-19 pandemic prevented us from undertaking further work on mAbs from dose 3 (**Fig. 2f, Fig. 3, and Supplementary Figs. 9-12**). 13-1C exhibited a similar footprint to 13-1F and more potent neutralizing activity than that of 13-1F, so it was selected for further investigation. The results revealed that 13-1C derived bsAbs, BI-2C1C and BI-1C2C, also exhibited more potent and broadly neutralizing activity against pseudoviruses than its parental mAbs and showed comparable neutralizing capacity to that of BI-2C5C and BI-5B2C (**Fig. 3e**). However, no matter 13-1C or BI-2C1C and BI-1C2C displayed much lower inhibitory activity against the authentic viruses (WT, Beta, Delta, BA.1, and BA.2) than that of 10-5B and its derived bsAbs in vitro (**Fig. 2f, Fig. 3f, and Supplementary Fig. 9**). Therefore, the protective efficacy and structural analysis of 13-1C and the bsAbs were not further examined in this work.

#4I. *There should be improved clarity regarding from which boost and subjects the neutralizing mAbs arose. Were these distributed across the cohort or from only a few subjects? Were these mostly from the 3rd boost? This should be clear as mAbs lose activity to variants finally resulting in only 3 with broad-variant activities.*

#4I Answer: Thank you for the suggestion. The neutralizing mAbs were isolated from 6 individuals who were sampled 1 month, 3 months and 6 months after the second

vaccine dose and 1 month after the third dose (lines 157-160, page 7). The detailed information has been added in Supplementary Table 4.

#4J. Please add information such as CDR3 sequence, SHM level, and variable gene for the 20 neutralizing antibodies to the table in Figure 2 for reference.

#4J Answer: Thank you for the suggestion. The characteristics of 20 neutralizing antibodies are described in Supplementary Table 4.

#4K. Figure 2e. Please indicate inside the figure which color means competition.

#4K Answer: Thank you for the suggestion. Figure 2e was revised with gradient color, and red indicates competition.

#4L. Line 196. The authors mentioned that antibodies isolated from samples collected after Dose 2 and Dose 3 had differential competition profiles and exhibited bias in epitope specificity. To further clarify this, the competitive ELISA with class I, II, III and IV antibodies should be performed on sera after the first, second, and third vaccine doses.

#4L Answer: Thank you for the suggestion. The competitive ELISA with class 1, 2, 3 and 4 antibodies has been performed on plasma after the first, second, and third vaccine doses according to the materials and methods of literature (**Fukushi S. Materials and methods Mol Biol. 2020, 2203:55-65.**). We found that plasma collected after Dose 3 had higher competitive antibody titers than samples collected after Dose 2 and Dose 1. However, no significant difference was observed in competitive antibody titers on these four classes at each time point (Fig. R2). Additionally, we analyzed the six samples, from which mAbs were isolated, to exclude the possible influence of selection bias. However, no skews in epitope specificity were observed from the plasma samples.

The discrepancy in competition profiles between neutralizing mAbs (Fig. 3e) and plasma might due to the impact of non-neutralizing antibodies in the plasma. Non-neutralizing antibodies comprised approximately 20-45% of the RBD-specific antibodies and displayed distinct competition features from the neutralizing antibodies in natural infection (**Wang, et al. Nature. 2021, 595: 426-431.**). In our work, non-neutralizing antibodies accounted for ~80% of the isolated mAbs and might exert their influence on the competitive results. In addition, this lack of concordance between the epitopes of plasma and mAbs is consistent with reports of other studies showing that the specificities of potent mAbs often do not recapitulate the plasma from which they were isolated (**Greaney, et al. Cell Host Microbe. 2021,29:463-476; Greaney, et al. Nat Commun. 2021,12:4196; Barnes, et al. Cell. 2020,182:828-842; Weisblum Y, Elife. 2020,9:e61312.**). Therefore, “antibodies” in the main text was revised to “neutralizing antibodies” (line 211, page 9).

Fig. R2 Plasma antibodies specific for different epitopes on SARS-CoV-2 RBD measured by competitive ELISA. Plasma collected after the first, second, and third vaccine doses were assayed for competitive titers with 4 structurally defined monoclonal antibodies using wild-type RBD protein as the coated antigen. CB6 was classified into Class 1, C121 (Class 2), COV2-2130 (Class 3), and COVA1-16 (Class 4). Statistical significance was determined by two-sided Kruskal Wallis test with subsequent Dunn's multiple comparisons.

#4M. As not all people are familiar with the various vaccines available, please provide a brief description of BBIBP-CorV (company, method of inactivation, geographical usage, etc.).

#4M Answer: Thank you for the suggestion. BBIBP-CorV was developed by Sinopharm's Beijing Institute of Biological Products (China) and became the first whole inactivated virus COVID-19 vaccine to receive emergency use authorization by the WHO. The HB02 strain with optimal replication and virus yields was selected and passaged in Vero cells to generate vaccine production. The BBIBP-CorV stock was inactivated by thoroughly mixing with β -propionolactone at a ratio of 1:4000 at 2–8°C. The vaccine was manufactured as a liquid formulation containing 4 μ g of total protein

with aluminium hydroxide adjuvant (0.45 mg/mL) per 0.5 mL. (**Wang, et al. Cell. 2020,182:713-721.**). BBIBP-CorV is approved in 93 countries, including China, Argentina, Mexico, Mongolia, Thailand, South Africa, Senegal, et al (<https://covid19.trackvaccines.org/vaccines/5/>). This information was added in the Methods section (**lines 515-522, page20**).

#4N. Line 100, grammar error, “were” not needed.

#4N Answer: Thank you for the reminder. The sentence was corrected to “The IgG responses to these viral structural proteins were much higher than those of IgM, with strong responses recorded after the two-dose regimen; a booster dose led to a further 2.3-9.4-fold increase in IgG titers against these antigens (Fig. 1b).” (**Lines 103-106, pages 5-6**).

REVIEWERS' COMMENTS

Reviewer #1 (Remarks to the Author):

Dear Editor,

The authors satisfactorily addressed all the issues I raised in the first review. In this regard, the paper can be published subject to your final decision.

Reviewer #2 (Remarks to the Author):

The reviewer has checked that the points raised have basically been corrected by the authors.

Reviewer #3 (Remarks to the Author):

The authors have sufficiently addressed my previous comments.

Minor comments:

It would be helpful to state in the text that Spike specific B cells were enriched for by cell sorting and that the higher SHM of control antibodies is thought to be from preexisting antibodies elicited by other coronaviruses that cross react with SARS-CoV-2 spike.

Figure 5. As depicted, it is not clear to my why the contacts for 6-2C differ for BA.1 and BA.4. The local environment (from sidechains shown) is identical. The loop with residues 370-375 appears to adopt an odd conformation in the middle panel. This may be due to low resolution or lower quality data. Please consider inspecting this region. Given this I am not sure how meaningful the table of contacts (5D) and the lengthy discussion of hydrogen bonds gained and lost are fully warranted.

Reviewer #4 (Remarks to the Author):

All concerns were adequately addressed.

Responses to Reviewer #3:

#3A. *It would be helpful to state in the text that Spike specific B cells were enriched for by cell sorting and that the higher SHM of control antibodies is thought to be from preexisting antibodies elicited by other coronaviruses that cross react with SARS-CoV-2 spike.*

#3A Answer: Thanks for your comments, and we recognize your concern. For the control antibodies, we isolated memory B cells of prevaccination samples of three participants in our cohort, without using SARS-CoV-2 spike or RBD as baits (**lines 165-170, page 8; lines 629-659, pages 24-25**). In general, very few SARS-CoV-2 spike- or RBD-binding memory B cells presented in healthy donors, and it's quite difficult to obtain enough specific memory B cells from limited human samples. Therefore, paired heavy and light chains of memory B cell receptors from healthy donors were used to represent baseline circulating memory B cell repertoire (*Liu, et al. Nature. 2020,584:450-456; Scheid, et al. Cell. 2021,184:3205-3221; Seow, et al. Cell Rep.2022,39:110757.*). The repertoire of memory B cell receptors encodes a comprehensive record of an individual's immunological encounters, usually experiencing continued development, and so have higher SHM levels than those of RBD-binding memory B cells that primed by the ongoing vaccination and without extensive germinal center experience (*Briney, et al. Nature, 2019,566:393-397.*). (**Lines 436-440, page 17**).

#3B. *Figure 5. As depicted, it is not clear to my why the contacts for 6-2C differ for BA.1 and BA.4. The local environment (from sidechains shown) is identical. The loop with residues 370-375 appears to adopt an odd conformation in the middle panel. This may be due to low resolution or lower quality data. Please consider inspecting this region. Given this I am not sure how meaningful the table of contacts (5D) and the lengthy discussion of hydrogen bonds gained and lost are fully warranted.*

#3B Answer: Thanks for your question. The local environment of BA.1 and BA.4 are slightly similar but not identical as BA.1 has S371L mutations while BA.4 exhibits S371F and T376A mutations in the spike (**Supplementary Fig. 17c**). The distinct loops of BA.1 (aa371-aa377, LAPFFTF) and BA.4 (aa371-aa377, FAPFFAF) exhibits main-chain structural variations (**Fig. R1a**), and the resulting side-chain conformational differences are also significant, evidenced by the different side-chain orientations of the F374 and F375 residues between BA.1 and BA.4 (**Fig. R1b**). We have carefully inspected the density map of this region and confirm the rationality of the models (**Fig. R1c**). We also compared our structures with prior reports by other groups (BA.1, PDB 7WHH; BA.4, PDB 7XNQ), which revealed similar results (**Fig. R1c**). Therefore, the structural differences in the local environment lead to the different contacts for antibody 6-2C with the spike of BA.1 and BA.4, and we believe that the table of contacts (**Fig. 5d**) and the discussion of hydrogen bonds gained and lost would be helpful to explain the mechanisms of binding and neutralization for antibody 6-2C.

a**b****BA.1 residues 372-377****c****BA.1 residues 371-377****BA.1 (PDB-7WHH) residues 371-377****BA.4 residues 372-377****BA.4 residues 371-377****BA.4 (PDB-7XNQ) residues 371-377****Fig. R1 The loop with residues 371-377 in the spike of Omicron BA.1 and BA.4.**

- (a) Cartoon diagram of 371-377 loop in BA.1 spike (salmon) superposed with that of BA.4 spike (purple).
- (b) Density map of BA.1 and BA.4 residues 372-377.
- (c) Side-chain conformation of residues 371-377 in BA.1 spike (salmon, this work; cyan, 7WHH) and BA.4 spike (purple, this work; yellow, 7XNQ).